# HiC-DC+ enables systematic 3D interaction calls and differential analysis for Hi-C and HiChIP

Merve Sahin [1,2], Wilfred Wong[1,2], Yingqian Zhan [3], Kinsey Van Deynze[4], Richard Koche [3] &
Christina S. Leslie [1✉]

Recent genome-wide chromosome conformation capture assays such as Hi-C and HiChIP have vastly expanded the resolution and throughput with which we can study 3D genomic architecture and function. Here, we present HiC-DC+, a software tool for Hi-C/HiChIP interaction calling and differential analysis using an efficient implementation of the HiC-DC statistical framework. HiC-DC+ integrates with popular preprocessing and visualization tools and includes topologically associating domain (TAD) and A/B compartment callers. We found that HiC-DC+ can more accurately identify enhancer-promoter interactions in H3K27ac HiChIP, as validated by CRISPRi-FlowFISH experiments, compared to existing methods. Differential HiC-DC+ analyses of published HiChIP and Hi-C data sets in settings of cellular differentiation and cohesin perturbation systematically and quantitatively recovers biological findings, including enhancer hubs, TAD aggregation, and the relationship between promoter-enhancer loop dynamics and gene expression changes. HiC-DC+ therefore provides a principled statistical analysis tool to empower genome-wide studies of 3D chromatin architecture and function.

[1] Computational and Systems Biology Program, Memorial Sloan Kettering Cancer Center, New York, NY, USA. [2] Tri-Institutional Training Program in Computational Biology and Medicine, New York, NY, USA. [3] Center for Epigenetics Research, Memorial Sloan Kettering Cancer Center, New York, NY, USA. [4] Bioinformatics Program, University of California at San Diego, San Diego, CA, USA. ✉email: cleslie@cbio.mskcc.org

Recent years have seen major improvements in genome-wide chromosome conformation capture technologies to facilitate the generation of high-quality, high-resolution maps of 3D genomic interactions. These developments include streamlined, low-input protocols and commercially available kits for Hi-C as well as alternative genome-wide assays such as Micro-C[1,2] and HiCAR[3] that improve recovery of shorter range and regulatory interactions. Meanwhile, multiple assays that enrich for specific classes of 3D interactions now allow high-resolution mapping of components of the 3D interactome with lower sequencing costs. In particular, HiChIP[4] couples Hi-C with chromatin immunoprecipitation for a protein or histone mark of interest, and capture Hi-C[5] uses an oligonucleotide probe library to enrich for interactions with specific loci. These advances in technology make possible the generation of high-resolution interaction data sets in multiple cell types, together with biological replicates, for studying changes in genomic architecture with cellular context and how 3D interaction changes are related to epigenomic and transcriptional changes.

However, high-resolution Hi-C, HiChIP, and other chromosome conformation capture data sets present many challenges for principled computational analysis and meaningful biological interpretation. First, these data sets are very large—~2B sequenced read pairs for "loop resolution" Hi-C maps and ~200M read pairs per replicate for HiChIP in human or mouse genomes —and therefore require considerable computational resources both for preprocessing to generate raw contact matrices and for downstream analyses. Second, while there are well-developed tools for visualization of Hi-C experiments (e.g., Juicebox[6] and the HiCExplorer Python tool[7]), there is no consensus on statistical methods for calling significant 3D interactions or for determining differential interactions between cell types, and published methods give highly discordant results. In particular, some well-established loop callers like HiCCUPS[8] are very conservative and best suited for identifying cohesin- and CTCF-mediated structural loops rather than regulatory interactions, while other methods are extremely permissive, as we have described previously[9]. Consequently, 3D genomic architecture studies sometimes lean on visualizations at specific loci at the expense of more comprehensive genome-wide analyses. Third, there are no "gold standard" data sets of validated 3D interactions, making it difficult to objectively benchmark Hi-C and HiChIP analysis tools. Finally, interpreting Hi-C data requires integrative analyses with 1D epigenomic data sets (chromatin accessibility, transcription factor, and histone mark ChIP-seq) and transcriptomic data that are informed by principles of gene regulation and chromatin biology, and these genome-wide integrative analyses remain challenging.

Here we present HiC-DC+ (Hi-C/HiChIP direct caller plus), a new Bioconductor package that enables principled statistical analysis of Hi-C and HiChIP data sets by building on the HiC-DC framework[9]. HiC-DC+ includes an efficient and parallelizable implementation of the HiC-DC background model to scale to high-resolution contact matrices and call significant 3D interactions in Hi-C, Hi-ChIP, and related assays. Additionally, HiC-DC+ performs differential analysis between conditions given replicate experiments using appropriate differential read count statistics by combining distance-dependent library scaling factors with DESeq2[10]. The package includes an implementation of A/B compartment and TAD callers to facilitate global integrative studies, and HiC-DC+ significant and differential interaction calls can be viewed with popular Hi-C visualization tools.

To address the difficulty of evaluating performance in the absence of gold standard benchmark data sets, we assembled a diverse collection of published Hi-C and HiChIP studies and defined biologically well-founded assessment metrics for method

comparison. First, we found that HiC-DC+ outperforms other HiChIP interaction callers for the task of identifying H3K27ac HiChIP promoter-anchored interactions as validated by CRISPRi-FlowFISH[11]. HiC-DC+ SMC1A HiChIP analysis also yields greater enrichment at subTAD boundaries than other methods. We next showed that differential Hi-C analysis by HiC-DC+ correctly recovers 3D architectural changes associated with knockout of the cohesin release factor WAPL[12], outperforming other differential tools at specific loci and in global assessments. Moreover, HiC-DC+ H3K27ac HiChIP differential analysis in mouse embryonic stem cells (mESCs) vs. mouse embryonic fibroblasts (MEFs) identifies experimentally validated mESC-specific enhancer hubs[13]. Finally, we analyzed very high coverage Hi-C data from a study of THP-1 monocyte to macrophage differentiation that reported modest changes in 3D loops[14]. Here, HiC-DC+ instead identifies widespread differential enhancer–promoter interactions that are associated with concordant gain or loss of H3K27ac-marked accessible elements and gene expression changes. Together, these analyses demonstrate the power of HiC-DC+ for systematic global analysis of 3D genomic interactions, including regulatory interactions, and biological interpretation.

## Results

**HiC-DC+ enables systematic analysis of Hi-C and HiChIP data.** HiC-DC+ estimates significant interactions in a Hi-C or HiChIP experiment directly from the raw contact matrix for each chromosome up to a specified genomic distance, binned by uniform genomic intervals or restriction enzyme (RE) fragments, by training a background model to account for random polymer ligation and systematic sources of read count variation. Similar to HiC-DC[9], HiC-DC+ uses negative binomial (NB) regression to estimate the expected read count in an interaction bin based on genomic distance and the GC content, mappability, and effective bin size based on RE sites in the corresponding pair of genomic intervals (Methods); genomic features derived from the set of RE recognition sites are provided as input to the model. As in HiC-DC and Fit-Hi-C[15], HiC-DC+ uses a two-step model fitting procedure to increase power (Methods). Given the higher coverage of current Hi-C and HiChIP experiments, by default HiC-DC+ does not use zero truncation in the background model, resulting in a somewhat simpler statistical model than in HiC-DC.

HiC-DC+ also enables differential analysis of Hi-C or HiChIP interactions between a pair of conditions, given replicate experiments. To do this, HiC-DC+ estimates genomic distance-dependent scaling factors from the data and uses DESeq2 to assess the significance of differential interactions (Methods). This approach is similar to the ACCOST method[16] but exploits existing capabilities of DESeq2 to perform the statistical test.

The HiC-DC+ Bioconductor R package features major improvements over HiC-DC. First, HiC-DC+ has greater time and memory efficiency (Supplementary Tables 1–5), which is required to process very deep (≥2B reads) data sets at high resolution. Second, HiC-DC+ supports standard Hi-C contact matrix file formats (.hic, .matrix, .allValidPairs) generated by HiC-Pro[7] or Juicer[17] as input (Fig. 1a). For convenience, the HiC-DC+ package also implements callers for topologically associating domains (TADs), through an implementation of the TopDom[18] algorithm, and A/B compartments[17]. The TAD caller can be run on contact matrices that have been normalized by HiC-DC+, namely observed/expected (O/E) counts or NB Z-score values (Fig. 1a, Methods), or by other methods such as ICE[19]. In addition, the HiC-DC+ package offers flexible modeling options. The dependence on genomic distance can

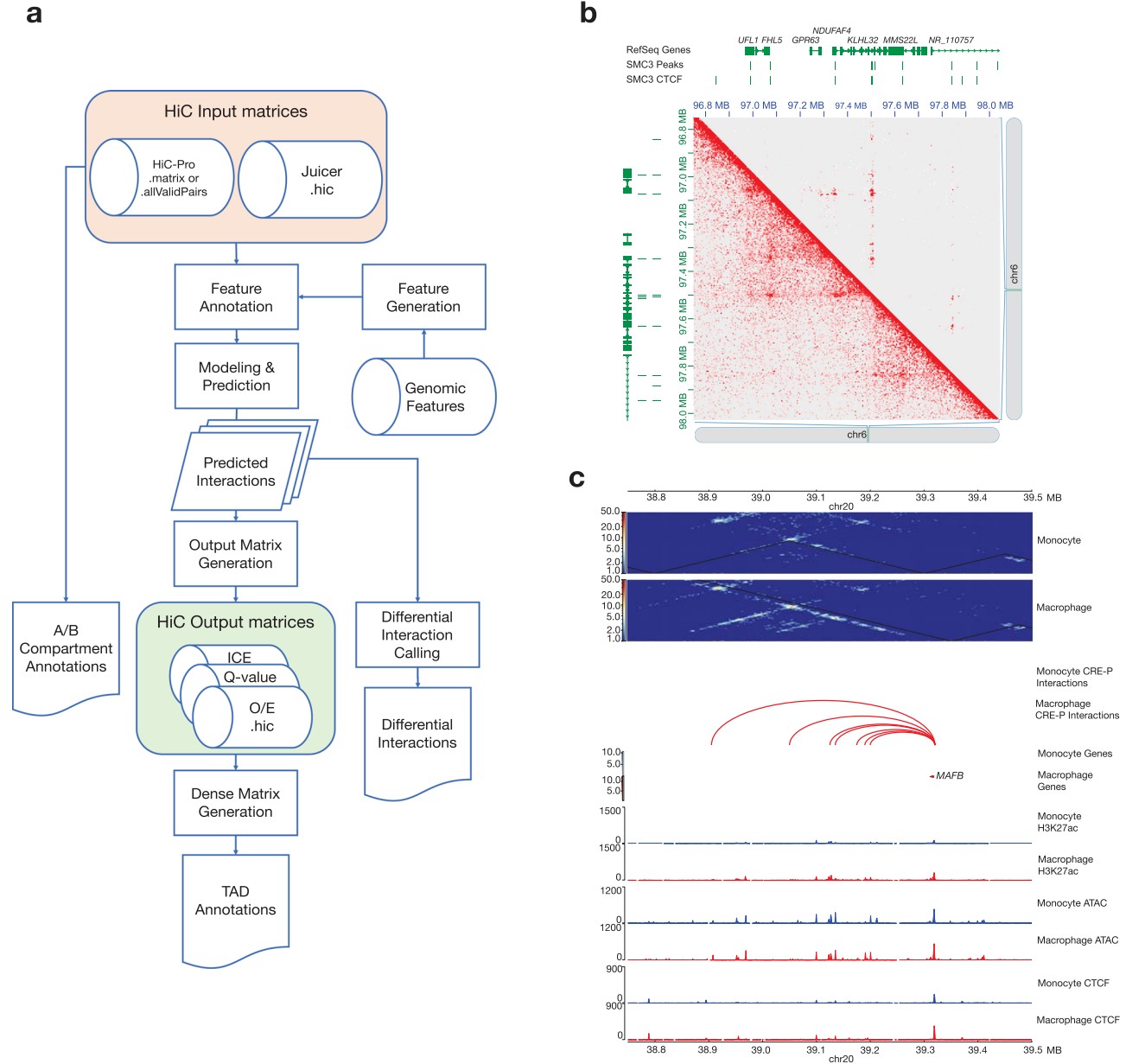

**Fig. 1 HiC-DC+ workflow and capabilities. a** Overview of the HiC-DC+ pipeline. **b** Visualization of HiC-DC+ results in Juicer: FDR-corrected *P* values estimated by HiC-DC+ shown in the upper triangle heatmap, KR normalized counts in the bottom triangle, for SMC1A HiChIP in GM12878 (merged replicates of GSM2138324, GSM2138325, GSM2138326, and GSM2138327) at 5 kb resolution. **c** Visualization of HiC-DC+ differential interactions in HiCExplorer: HiC-DC+ *Z*-score normalized Hi-C counts in THP-1 monocytes and macrophages (PRJNA385337) at 5 kb resolution are shown as heatmaps, along with differential Hi-C interactions as arcs and H3K27ac (GSM2544236, GSM2544237, GSM2544238, and GSM2544239) and CTCF (GSM2544244, GSM2544245, GSM2544246, and GSM2544247) ChIP-seq and ATAC-seq (GSM2544216, GSM2544217, GSM2544220, GSM2544221, GSM2544224, GSM2544225, GSM2544228, GSM2544229, GSM2544218, GSM2544219, GSM2544222, GSM2544223, GSM2544226, GSM2544227, GSM2544230, and GSM2544231) as signal tracks for a region encompassing the gene *MAFB*.

either be defined using splines as in HiC-DC or using power decay, covariates can be easily removed or new covariates added, and the background model for counts can be estimated using either hurdle or NB regression. Finally, in addition to the default constant dispersion model, HiC-DC+ implements a variable distance-dependent dispersion model within the GLM framework. While our benchmarking focuses on Hi-C and HiChIP, the flexibility of HiC-DC+ potentially enables applications to diverse chromatin capture data sets. For example, the variable dispersion option appeared to be useful for Micro-C, where the dispersion highly varies with distance, while the hurdle regression option seemed helpful for ChIA-PET (Supplementary Fig. 1a).

To visualize significant HiC-DC+ 3D interactions and differential loops, results from the package (FDR-corrected *P* values, or O/E or *Z*-score normalized counts in .hic file format) can be readily supplied to popular visualization tools such as Juicer[17] and HiCExplorer[7]. For example, Fig. 1b shows the results of HiC-DC+ analysis of a SMC1A HiChIP data set in the GM12878 lymphoblastoid cell line[4,8] in a 1.2 Mb region of chr 6 using Juicer, with KR normalized count data below the diagonal and significant interactions at FDR < 0.05 above the diagonal. HiCExplorer is useful for visualizing HiC-DC+ normalized contact matrices in triangular format and significant interactions as arcs, alongside other genomic tracks. In Fig. 1c, HiC-DC+

Z-score normalized Hi-C data in untreated and PMA-treated THP-1 cells[14] are shown at a 1 Mb region encompassing *MAFB*, an important regulator of monocyte to macrophage differentiation, together with HiC-DC+ differentially gained enhancer–promoter interactions and signal tracks for ATAC-seq, H3K27ac, and CTCF ChIP-seq.

We used interactions called from pooled replicates for most of our benchmarking case studies; however, we also tested the robustness of our significant interactions. In the THP-1 Hi-C data set as well as in mESC H3K27ac HiChIP data[4], scatter plots of $-\log_{10}(P)$ and Z-scores for each pair of replicates show that the significance of interactions is in high agreement between replicates, suggesting strong reproducibility of results for Hi-C and HiChIP (Supplementary Fig. 2). We also checked the consistency of HiC-DC+ interactions at different bin resolutions and found ~80% of interactions called at 5, 25, and 50 kb also as significant at 10, 50, and 100 kb, respectively, in THP-1 monocytes and macrophages (Supplementary Fig. 3).

**HiC-DC+ detects enhancer–promoter interactions from HiChIP.** HiChIP enriches for 3D interactions associated with a protein or histone modification of interest by applying ChIP to the contact library after nuclear lysis[4]. The HiC-DC+ statistical model proved to be well suited for HiChIP analysis without requiring any additional covariates, such as signal from a parallel ChIP-seq experiment or HiChIP self-ligation read counts, which have been used to mimic the ChIP-seq signal[4]. Since HiC-DC+ covariates such as GC content and mappability account for sources of systematic biases in high-throughput sequencing, we may already be implicitly modeling a component of ChIP bias. By contrast, some existing HiChIP analysis methods like hichipper[20] and FitHiChIP[21] indeed attempt to call genomic peaks directly from HiChIP or use a parallel ChIP-seq experiment as a step in their modeling.

To benchmark HiC-DC+ on HiChIP, we ran the model on existing H3K27ac HiChIP in K562 cells[4] at 5 kb resolution and assessed using recently generated CRISPRi-FlowFISH data for 22 genes[11] (Methods). CRISPRi-FlowFISH quantifies enhancer function in terms of the estimated log fold change of target gene expression due to enhancer inactivation by using a pooled CRISPRi screen against candidate enhancers and sorting cells using RNA fluorescence in situ hybridization (FISH) against the target gene. We reasoned that a successful H3K27ac HiChIP analysis pipeline should identify significant loops between the target gene promoter and FlowFISH-validated accessible sites, and rank these loops above interactions between the target promoter and accessible sites with insignificant FlowFISH effect.

Therefore, we ranked candidate interactions at 5 kb between the target promoter and each tested regulatory element by HiC-DC+ *P* value. Similarly, we ranked candidate interactions using three other published methods: FitHiChIP[21], an adaptation of Fit-Hi-C for HiChIP (using the loose background model(L)); MAPS[22], another approach based on fitting a generalized linear model to estimate a background distribution; and hichipper[20], a method that models the background read density in terms of proximity to RE sites. When we evaluated performance per target gene by auPR (area under the precision-recall curve), HiC-DC+ outperformed hichipper and MAPS (Fig. 2a, *P* < 0.01, signed rank test). Although per gene auPR was not significantly different between HiC-DC+ and FitHiChIP (*P* = 0.15, signed rank test), HiC-DC+ had a higher median per gene auPR (0.307 for HiC-DC+ vs 0.245 for FitHiChIP); HiC-DC+ also significantly outperformed other model variants of FitHiChIP (Supplementary Fig. 4, *P* < 0.01, signed rank test). Moreover, when we examined the predicted interactions at FDR < 0.01 across all genes for each

method, we found that HiC-DC+ was most successful in identifying promoter–enhancer loops of enhancers with large effect size on target gene expression as assessed by CRISPRi-FlowFISH (Fig. 2b, *P* < 0.011 for FitHiChIP, *P* < 0.005 for MAPS, and *P* < 0.001 for hichipper, rank sum test).

While FitHiChIP was the runner-up in terms of overall auPR (Supplementary Fig. 5a, 0.180 for FitHiChIP vs. 0.205 for HiC-DC+), suggesting a reasonable ranking of promoter-to-candidate–enhancer interactions, it called many more interactions than HiC-DC+ at a fixed FDR <0.01 threshold, suggesting that its *P* values may be somewhat inflated. Indeed, at the *PLP2* and *FTL* loci (Fig. 2c, d), FitHiChIP identifies both positive and negative enhancers but generates a large number of false positive interactions, whereas HiC-DC+ identifies significant interactions that coincide with positive enhancers and H3K27ac peaks. We further compared the performance of HiC-DC+ and FitHiChIP on mESC H3K27ac HiChIP[4] based on CRISPRi-validated enhancer–promoter pairs in mESC[11] and found that HiC-DC+ was more successful in identifying promoter–enhancer loops of enhancers with large effect size on target gene expression (Supplementary Fig. 6, overall auPR of 0.54 for HiC-DC+ vs. 0.29 for FitHiChIP(L)).

We also generated aggregate peak analysis (APA) plots using HiChIP interactions unique to HiC-DC+ and to FitHiChIP and found that interactions exclusive to HiC-DC+ had higher APA scores (Methods, Supplementary Fig. 7, 1.28 vs. 1.16 for K562 H3K27ac HiChIP, 3.29 vs. 1.45 for mESC H3K27ac HiChIP).

Finally, we sought to directly address the issue of whether it is preferable to normalize for ChIP-seq bias in HiChIP analysis, or conversely whether explicitly incorporating ChIP signal in the model normalizes away the true signal and decreases power. At least in the context of H3K27ac HiChIP analysis for the detection of functional regulatory interactions, we found the latter to be the case. We incorporated parallel H3K27ac ChIP-seq signal in K562 cells either as a continuous or a categorical covariate in the HiC-DC+ model in several ways, analogous to other methods (Supplementary Note 1). In all cases, we found that normalization for ChIP bias decreased performance for detection of functional promoter–enhancer interactions as validated by CRISPRi-FlowFISH (Supplementary Fig. 8). Moreover, the performance advantage of HiC-DC+ over FitHiChIP cannot be attributed to stronger H3K27ac ChIP enrichment in HiC-DC+ HiChIP interactions. Indeed, for high confidence interactions (recall ≤ 2) of each method in the CRISPRi-FlowFISH candidate set, there was no significant difference in ChIP signal between HiC-DC+ and FitHiChIP, and for lower confidence interactions (recall > 0.2) where FitHiChIP outperforms HiC-DC+ in precision (Supplementary Fig. 5), FitHiChIP interactions overlapping the candidate pairs actually had slightly but significantly higher ChIP signal than HiC-DC+ interactions (Supplementary Fig. 9). Furthermore, we found that top 100,000 interactions identified by HiC-DC+ had lower average normalized ChIP signal levels in both anchors compared to FitHiChIP (Supplementary Fig. 10).

Despite evidence that promoter–enhancer interactions identified by CRISPRi-FlowFISH are enriched for shorter range interactions that are detectable by H3K27ac signal and accessibility alone (Supplementary Note 2 and Supplementary Fig. 11), enhancer screening data provides a useful benchmark data set for comparing HiChIP interaction callers.

**HiC-DC+ accurately identifies cohesin-mediated loops.** As a second test case, we benchmarked HiC-DC+ and existing HiChIP interaction callers on published HiChIP data for SMC1A, a subunit of the cohesin complex, in GM12878[4]. The other HiChIP interaction calling methods identified vastly different

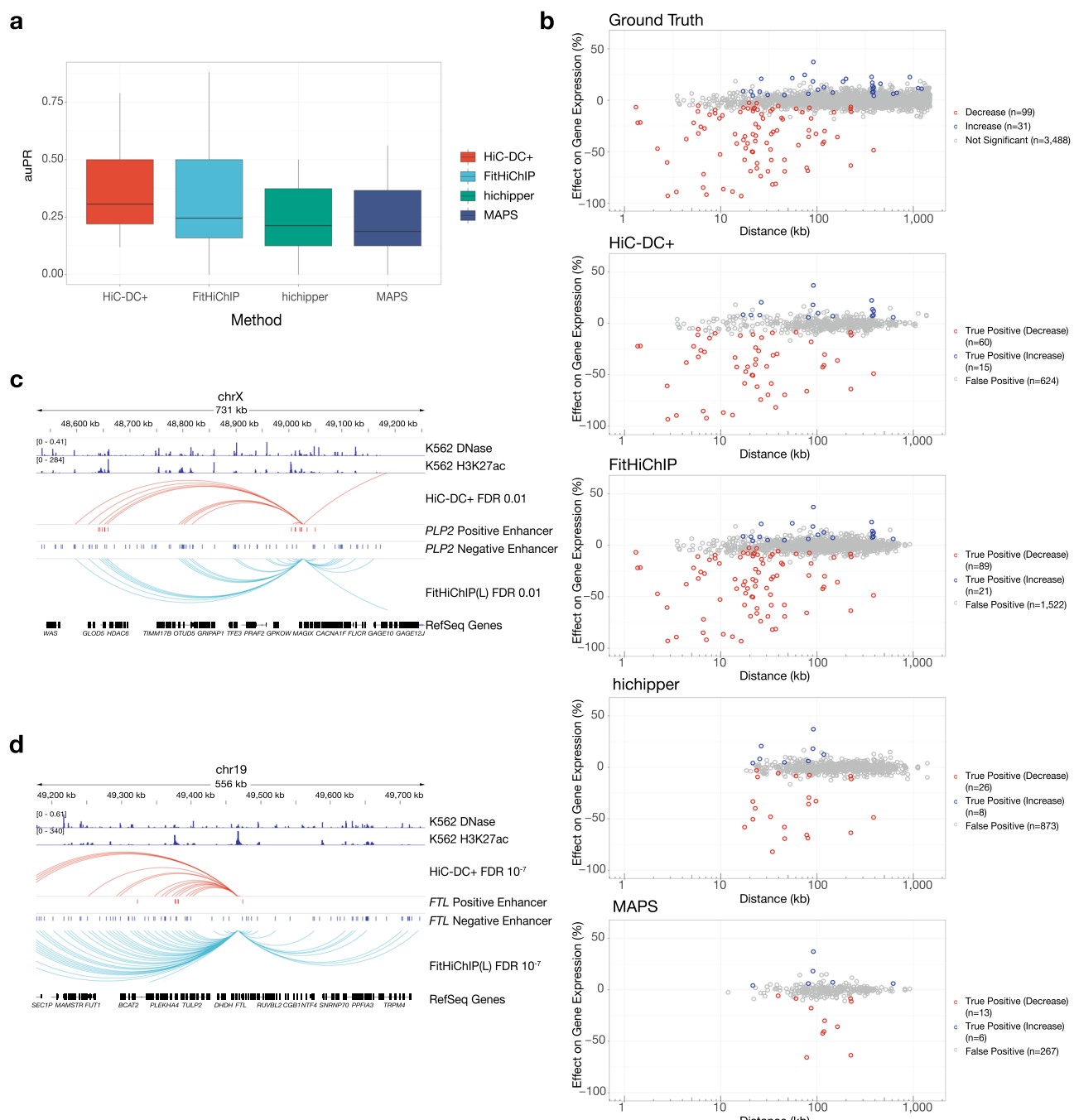

**Fig. 2 HiC-DC+ accurately identifies enhancer–promoter interactions from H3K27ac HiChIP. a** Evaluation of method performance by per gene auPR values. For each target gene, we ranked the candidate regulatory elements tested by CRISPRi-FlowFISH in K562 cells[11] based on the significance ($P$ value) of corresponding 5 kb HiChIP interactions (merged replicates of GSM2705043, GSM2705044, and GSM2705045) with the target promoter as estimated by each method ($n = 22$ genes; Methods). Centers of the boxes indicate median values, the lower and upper hinges correspond to the first and third quartiles and the upper (lower) whiskers extend from the hinge to the largest (smallest) value no further than 1.5 times the distance between the first and third quartiles. **b** Comparison of performance for identifying promoter–enhancer loops with large effect sizes on target gene expression as assessed by CRISPRi-FlowFISH. Each dot in the scatterplots represents one tested promoter–enhancer pair: true positives resulting in decreased target gene expression upon enhancer inactivation are shown in red, true positives resulting in increased target gene expression shown in blue, false positives in gray. **c** Promoter–enhancer loops identified by HiC-DC+ and FitHiChIP (FDR <0.01) are shown as arcs for *PLP2* gene, along with all candidate enhancers tested by CRISPRi-FlowFISH[11] and signal tracks for K562 DNase-seq (GSM816655) and H3K27ac ChIP-seq (GSM733656). **d** Promoter–enhancer loops identified by HiC-DC+ and FitHiChIP (FDR < $10^{-7}$) are shown as arcs for *FTL* gene, along with all candidate enhancers tested by CRISPRi-FlowFISH and K562 DNase-seq and H3K27ac ChIP-seq signal tracks.

numbers of significant cohesin-mediated loops at 5 kb resolution with genome-wide FDR < 0.01 (Supplementary Table 6). When we restricted to loops whose anchors both contained CTCF motifs, a majority of such loops (75–84%) for all methods except hichipper (49%) were associated with a pair of convergently oriented motifs (Supplementary Fig. 12). We also examined the enrichment of significant interactions from each method at the boundaries of previously reported subTADs in GM12878 based on Hi-C analysis[8]. We found that HiC-DC+ interactions were most highly enriched at subTAD boundaries, with FitHiChIP as runner-up (Supplementary Figs. 13, 14). We alternatively used HiCCUPS loops on Hi-C data in GM12878 as the ground truth and found that detection of these loops by SMC1A HiChIP in GM12878 without modeling ChIP bias gives very similar performance to FitHiChIP with modeling ChIP bias (Supplementary Fig. 15).

**Differential Hi-C analysis recovers TAD aggregation**. To call differential Hi-C and HiChIP interactions between a pair of conditions, we estimate distance dependent DESeq2 size factors so that the median normalized count for pairs of bins at each given distance in each matrix is the same (Methods). We first pool all replicates to call significant interactions for each cell type and assemble an atlas of significant 3D interactions, and then we use count data from replicates in DESeq2 to find differential interactions. Dispersion and MA plots behave as expected, suggesting that our normalization works appropriately (Supplementary Fig. 16). We tested the reproducibility of HiC-DC+ differential interactions using Hi-C data with four replicates in mESC and neural precursor cells (NPC)[23] and called differential interactions between mESC and NPC using all three replicates (123), and all pairwise combinations of replicates (12, 13, and 23). We found that only about 10% of differential interactions called using two replicates at a time are exclusive to individual pairs, suggesting high reproducibility of differential interactions (Supplementary Fig. 17). As expected, using all three replicates gave more power to detect interactions than any pair of replicates.

To benchmark our differential Hi-C interactions against diffHic[24], multiHiCcompare[25], and Selfish[26], we ran HiC-DC+ and other tools on Hi-C data in HAP1 and WAPL knockout HAP1 cells at 25 kb resolution[12]. Haarhuis et al. showed that removing WAPL affects chromosome topology on a global scale through the formation of longer loops and strongly increased interaction frequencies between nearby TADs[12]. In particular, they highlighted three genomic regions to show accumulation of contacts at TAD corners (Fig. 3a, top row) and loop extension linked to TAD aggregation (Fig. 3a, middle and bottom rows) upon WAPL loss. HiC-DC+ can detect these enriched loops more effectively than other methods at FDR <0.05. Other methods also call at least two times more significant interactions than HiC-DC+, suggesting P value inflation (Supplementary Fig. 18). The loop lengths of differential interactions with HiC-DC+ span a wide range, while other methods appear biased towards short- or long-range interactions (Supplementary Fig. 19). In addition, differential interactions that are unique to HiC-DC+ have higher APA scores than those unique to other methods tested, suggesting that HiC-DC+ differential interactions are highly enriched in Hi-C contacts (Methods, Supplementary Figs. 20, 21, and 22).

We also performed contact frequency analysis between each TAD and its five flanking TADs to assess whether differential interaction analysis can reveal the TAD aggregation upon WAPL deficiency. We plotted the number of differential interactions identified by each method at FDR < 0.05 that are lost or gained in

ΔWAPL vs wild type (Fig. 3b). Haarhuis et al. claimed that WAPL deficiency mediated gains in interactions at the corners of TADs and loss of intra-TAD interactions. Of the methods tested, HiC-DC+ and Selfish recovered both findings: a majority of gained interactions connected neighboring TADs together with a drastic loss of intra-TAD interactions. However, Selfish was not able to find differential interactions at the three highlighted loci in Fig. 3a.

**Differential HiChIP analysis identifies enhancer hubs**. To evaluate differential HiChIP interactions, we examined H3K27ac HiChIP data from Di Giammartino et al.[13] This study mapped high-resolution regulatory loops in MEF and mESC and discovered enhancer hubs, defined as sets of highly connected enhancers interacting with cell-type-specific genes. Further, they disrupted KLF binding motifs within mESC enhancer hubs using CRISPR-Cas9 to test the hypothesis that enhancer hubs coordinate the expression of pluripotency-associated genes. Inactivation of enhancer hubs resulted in coordinated downregulation of all connected genes without affecting the neighboring non-hub genes in mESC. We used HiC-DC+ differential HiChIP analysis to identify mESC-specific enhancer hubs that are lost in MEF. HiC-DC+ detected mESC-specific interactions of the *Tbx3* enhancer hub with *Aw545942*, *Gm16063*, and *Tbx3* genes, all validated target genes; consistent with validation experiments, HiC-DC+ did not find a differential interaction with *Med13l*, whose expression is not affected by inactivation of this hub (Fig. 3c). HiC-DC+ also captured increased interaction of *Zfp42* enhancer hub with *Zfp42* and *Triml2* genes in mESC (Supplementary Fig. 23). Notably, another differential HiChIP interaction tool, diffloop[27], could not detect any differential interactions comprising these two hubs.

**Differential HiChIP interactions are not due to ChIP bias**. One concern that has been raised in differential HiChIP analysis is whether detected cell-type specific events can arise due to differential ChIP signal alone rather than a true change in the underlying 3D interaction[21]. Two problematic scenarios have been proposed: first, that the differential HiChIP interaction might arise purely due to bias from differential ChIP signal in one cell type; or second, that there is a stable 3D interaction across both cell types with a gain or loss in ChIP signal, altering the biological interpretation of the event. These scenarios have been put forward as motivation to use parallel ChIP-seq signal, or HiChIP self-ligation signal as a proxy for ChIP, to normalize or at least annotate HiChIP interactions[20,21].

To address these concerns, at least in the context of regulatory interactions and H3K27ac HiChIP analysis, we analyzed HiC-DC+ differential analysis results for two ENCODE cell lines, GM12878 and K562) in the context of parallel Hi-C and H3K27ac ChIP-seq data (Supplementary Note 3).

We found that a majority of all differential HiChIP interactions called by HiC-DC+—with no normalization for ChIP—were in fact supported by a concordant change in Hi-C signal (58%, n = 269,995) or in ChIP-seq signal at one or both anchors (70%, n = 327,437) or concordant changes in both Hi-C and ChIP-seq signals (42%, n = 199,029) (Supplementary Fig. 24). Most of the differential HiChIP interactions with a concordant change in ChIP-seq signal also exhibit a concordant change in Hi-C signal (61%), suggesting that the gain/loss of the ChIP signal most often happens simultaneously with change in the underlying 3D interaction, confounding the notion of normalizing for ChIP bias in HiChIP analysis.

We also identified differential HiChIP interactions with concordant differential ChIP signal in one of the anchors but

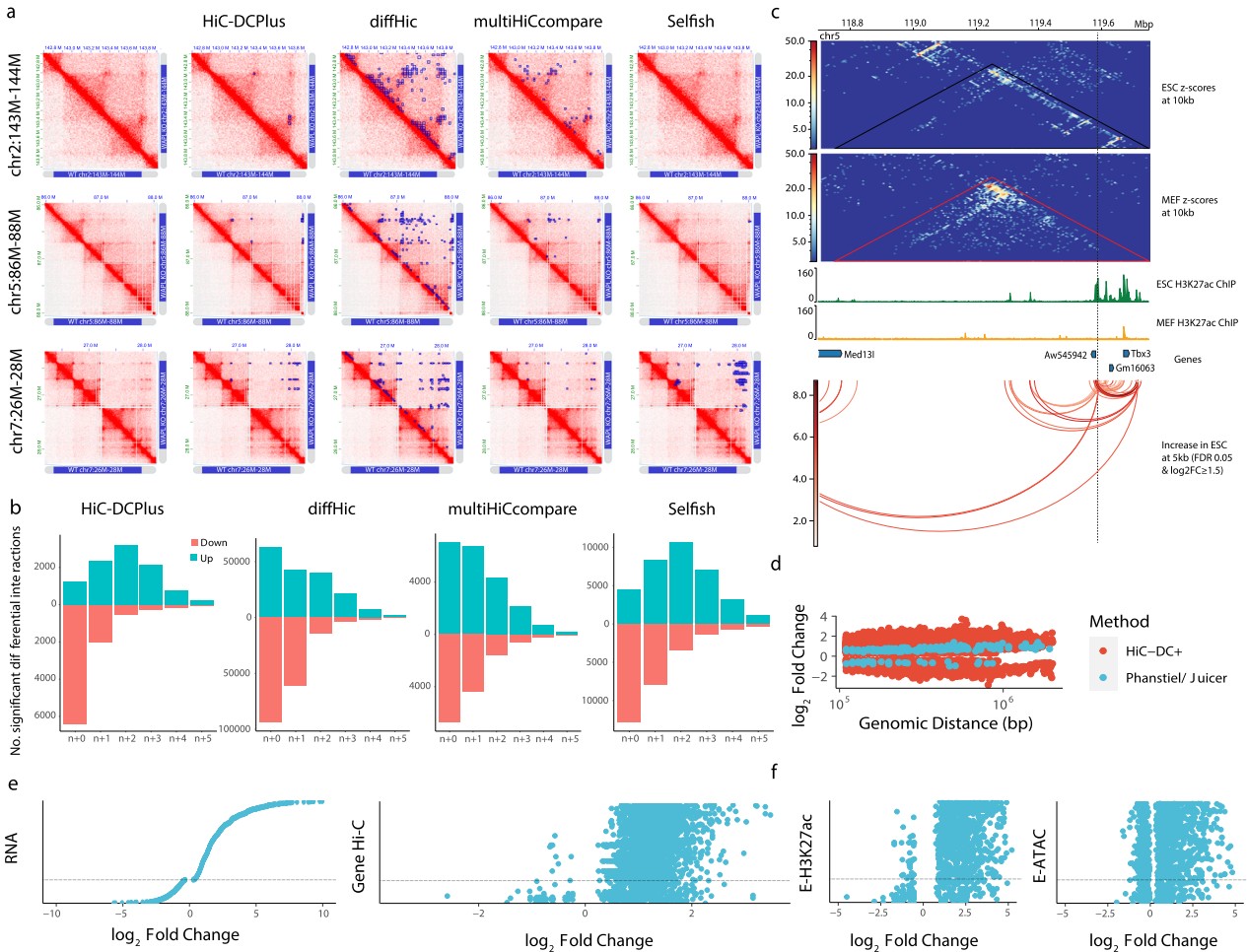

**Fig. 3 HiC-DC+ differential Hi-C and HiChIP analysis validates global chromatin folding changes during cohesin perturbation and cellular differentiation. a** ICE-normalized Hi-C counts for HAP1 WT (lower triangle) (merged replicates of GSE74072) and HAP1 WAPL knockout (upper triangle) (merged replicates of GSM2515800, GSM2515801, and GSM2515802) at 10 kb resolution were visualized for three genomic regions highlighted in Haarhuis et al., 2017[12]. For each of these regions, differential Hi-C interactions between *ΔWAPL* knockout vs. WT cells identified by each method at 25 kb resolution are shown as blue squares. **b** Differential interaction analysis between each TAD and its five flanking TADs. The barplots show the number of differential interactions in *ΔWAPL* vs wild type belonging to each category as identified by each method at FDR < 0.05. **c** HiC-DC+ mESC-specific H3K27ac HiChIP (replicates of GSM3103921, GSM3103922, GSM3103923, and GSM3103924) interactions recover the *Tbx3* enhancer hub with *Aw545942*, *Gm16063*, and *Tbx3* genes at 5 kb resolution, all validated targets from Di Giammartino et al., 2019. Black and red triangles show the TADs called in mESC Hi-C (merged replicates of GSM2533818, GSM2533819, GSM2533820, and GSM2533821). **d** Distribution of log fold changes of statistically significant looping events identified by HiC-DC+ and modified Juicer (Phanstiel et al., 2017) as a function of genomic distance (log scale). **e** Log fold changes of Hi-C interactions (PJNA385337) and gene expression (GSM2599707, GSM2599708, GSM2599709, and GSM2599710) between THP-1 macrophages vs. monocytes. Each dot in the Hi-C plot represents an interaction between the promoter of a differentially expressed gene and a putative enhancer (Methods), while each dot in the RNA-seq plot represents a differentially expressed gene. **f**. Log fold changes of H3K27ac ChIP-seq (GSM2544236, GSM2544237, GSM2544238, and GSM2544239) and ATAC-seq (GSM2544216, GSM2544217, GSM2544220, GSM2544221, GSM2544224, GSM2544225, GSM2544228, GSM2544229, GSM2544218, GSM2544219, GSM2544222, GSM2544223, GSM2544226, GSM2544227, GSM2544230, and GSM2544231) signal between THP-1 macrophages vs. monocytes at enhancer anchors of the differential promoter–enhancer loops identified by HiC-DC+ at 5 kb resolution. Each dot represents a putative enhancer.

only a small change in Hi-C signal (27%, $n = 128,408$ of all differential HiChIP interactions). We hypothesized that these differential interactions included real enhancer–promoter interactions that are not captured with high sensitivity in Hi-C. Indeed, we found that differential HiChIP interactions linked to a promoter in one anchor with small changes in Hi-C but substantial changes in ChIP intensity in the other anchor are depleted in significant Hi-C interactions (odds ratio = 0.71, $P < 1.0 \times 10^{-16}$, Fisher's exact test) but enriched in differentially expressed genes (odds ratio = 1.11, $P < 7 \times 10^{-3}$, Fisher's exact test) compared to other promoter-anchored HiChIP interactions, suggesting that Hi-C lacked the sensitivity to detect these

promoter-anchored interactions in either cell type, at least at the sequencing depth of the available experiments (Supplementary Fig. 25).

Therefore, we did not find strong evidence of differential HiChIP interactions arising as an artifact due to differential ChIP bias, or even of differential HiChIP interactions associated with a stable 3D interaction and differential ChIP. While we cannot exclude these possibilities, our systematic analysis suggests that HiChIP normalization by ChIP-seq may not be helpful and supports the validity of HiC-DC+ differential HiChIP interactions, at least in the context of H3K27ac HiChIP analysis.

**Differential HiC-DC+ interactions link to expression changes.**
Finally, to demonstrate the value of HiC-DC+'s rigorous differential analysis, we reanalyzed very high resolution in situ Hi-C data from the THP-1 cell line model of monocyte to macrophage differentiation, together with epigenomic and transcriptomic data sets, to relate chromatin dynamics to gene expression changes[14].

The original study used Juicer (HiCCUPS) to call interactions and adapted DESeq2 for differential interaction analysis but identified only 217 differential looping events genome-wide at a 10 kb resolution (nominal $P < 0.001$). This modest number led the authors to propose a model where macrophage-specific genes are upregulated through activation of preexisting DNA loops together with a smaller number of gained loops. By contrast, HiC-DC+ identified 64,844 differential interaction events at 10 kb resolution (adjusted $P < 0.01$). Of the original differential loop calls, 169 were also called by HiC-DC+ (FDR < 0.01) and 147 of these were found to be differential (adjusted $P < 0.01$). Indeed, we found that HiC-DC+ identified more differential interaction events at every genomic distance, and these differential interactions exhibited stronger effect sizes than those in the original study (Fig. 3d). These results suggest that Juicer is not only too conservative in calling loops but also preferentially identifies loops that exhibit little dynamic behavior. Therefore, we revisited the association of regulatory loop dynamics and gene expression changes.

We reran HiC-DC+ at 5 kb resolution in order to more precisely identify potentially regulatory interactions and found 25,691 differential events (adjusted $P < 0.05$). We next considered differentially expressed genes between THP-1 monocyte and macrophages (FDR < 0.05) and found that 1673 (out of 6782) had at least one promoter-anchored differential interaction. Strikingly, upregulated genes were associated with gained/strengthened promoter-anchored interactions, while downregulated genes were associated with lost/weakened promoter-anchored interactions. Moreover, differential promoter-anchored interactions at upregulated genes tended to be shorter range (median genomic distance 135 kb), while those at downregulated genes were longer range (Supplementary Fig. 26a, b median genomic distance 185 kb). GO enrichment for genes with differential promoter-anchored interactions identified functional annotations associated to monocyte to macrophage differentiation (Supplementary Fig. 27). These results all suggest that gain or loss of promoter-anchored interactions lead respectively to up- or downregulation of gene expression in monocyte to macrophage differentiation, presumably through differential connections to enhancers.

To investigate whether differential promoter-anchored interactions indeed include promoter–enhancer events, we next constructed an atlas of enhancers using ATAC-seq peaks that overlap with H3K27ac ChIP-seq peaks as a proxy for active enhancer elements (Methods). We then restricted analysis to looping events between enhancers in the atlas and promoters (Fig. 3e). Upregulated genes were associated with gained/strengthened promoter–enhancer interactions, while almost all lost/weakened promoter–enhancer interactions were associated with genes with no significant expression change or those that were significantly downregulated. Further, H3K27ac ChIP-seq and ATAC-seq changes at the interacting enhancer elements correlated with expression changes of the target gene, while weaker changes in ATAC and H3K27ac signal were observed at the promoter (Fig. 3e, f and Supplementary Fig. 28). Interestingly, CTCF changes at those promoter anchors were strongly associated with the changes in gene expression, although the enhancer sites displayed at best weakly concordant changes in ATAC-seq or CTCF ChIP-seq signal (Supplementary Fig. 28). Phanstiel et al. identified multiple gained loops at the *MAFB* locus, and HiC-DC+ analysis also found many enhancer–promoter interactions

for *MAFB* (Fig. 1c). We also recovered additional genes with previously unidentified differential looping events such as *GALNT4*, a gene involved in the mucin glycosylation pathway (Supplementary Fig. 29).

## Discussion

We have shown that HiC-DC+ facilitates genome-wide analyses of Hi-C and HiChIP data sets and integration with 1D epigenomic and transcriptomic data, recovering previous biological findings in a systematic and global fashion while enabling new insights on the role of 3D genomic architecture in gene regulation. Therefore, HiC-DC+ addresses an important gap in currently available tools for 3D interaction calls and differential analysis and can enable advances in genome-wide understanding of chromatin architecture and dynamics. As the HiC-DC+ package offers flexibility in feature and statistical model definition (options to use NB or hurdle regression, constant or distance-dependent variable dispersion) along with substantially improved efficiency, it has the potential to be applied to a wide range of chromatin capture data sets. For example, we tested HiC-DC+ on Micro-C and CTCF ChIA-PET data sets utilizing NB regression with variable dispersion at 5 kb for Micro-C and hurdle regression at 10 kb for ChIA-PET. Our initial analyses show high overlap with HiCCUPS loops for both CTCF ChIA-PET and Micro-C, suggesting promising results for future exploration (Supplementary Fig. 1).

One contribution of our study is the development of benchmark datasets and analysis tasks to assess the performance of significant and differential 3D interaction callers. For example, we used enhancer screening results from CRISPRi-FlowFISH and individual CRISPRi experiments to assess the performance of H3K27ac HiChIP interaction calls in K562 and mES cells. This analysis confirmed that HiC-DC+ outperforms other HiChIP analysis methods for identifying functional enhancer–promoter interactions, as validated by CRISPRi. One caveat is that the set of "positive" enhancer–promoter interactions are those for which CRISPRi perturbation of the enhancer leads to a significant change in expression of the target gene. It is possible that there are true physical H3K27ac-associated 3D enhancer–promoter interactions that are not detected by CRISPRi due to lack of detectable impact on target expression, for example due to buffering effects of additional interactions involving the target promoter. Another limitation of existing CRISPRi-FlowFISH is the relative paucity of validated long-range enhancer–promoter interactions among the screened elements. Finally, when comparing against HiChIP methods that correct for ChIP-seq bias, there is the confounding issue that strong H3K27ac ChIP-seq signal at accessible elements is associated with true functional enhancer–promoter interactions with detectable effect size in CRISPRi-FlowFISH, especially at close proximity to the promoter, as shown by activity-by-contact and activity-by-distance analysis (Supplementary Fig. 11 and Supplementary Note 2). This raises the possibility that HiC-DC+, by not normalizing for ChIP-seq signal, is implicitly using ChIP bias to perform well in the CRISPRi-FlowFISH task. However, there is in fact no significant difference in ChIP signal between the high confidence interactions predicted by HiC-DC+ vs. FitHiChIP, a method that does correct for ChIP bias (Supplementary Fig. 9). Since the goal of H3K27ac HiChIP is to enrich for regulatory interactions, including enhancer–promoter interactions that are expected to influence target gene regulation, we believe CRISPRi-FlowFISH is a useful benchmark data set for HiChIP interaction callers, despite the limitations described here.

Many interactions detected in H3K27ac HiChIP data are putative enhancer–enhancer interactions, for which there is no

gold standard data set for evaluation. Higher-resolution assays like HiCAR and Micro-C appear to enrich for shorter range regulatory interactions, so a careful analysis of these chromatin conformation assays together with parallel H3K27ac ChIP-seq could establish a curated set of enhancer–enhancer interactions on which to benchmark H3K27ac HiChIP calls. One could also try to validate HiChIP-based interactions involving specific enhancers using independent 4C experiments with the enhancer locus as viewpoint. Still, such analyses depend on statistical analysis of high-throughput 3C-based experiments and are therefore not entirely orthogonal to HiChIP. Moreover, we would ultimately want a functional validation for enhancer–enhancer interactions, and for this we would need a better understanding of their regulatory role. For example, one might perform CRISPRi experiments that target one enhancer participating in the putative enhancer–enhancer interaction and perform H3K27ac qPCR at the other enhancer to assess if there is distal loss of activity. Data sets of this nature may emerge in the next few years.

The issue of whether and how to normalize for chromatin IP signal in HiChIP experiments—either using parallel ChIP-seq data or self-ligation reads as a surrogate for ChIP[20,21]—is not resolved in the literature. The concern often raised is that without normalization, HiChIP interactions may be biased for higher ChIP enrichment, and differential HiChIP interactions may be due to differential ChIP enrichment rather than an underlying 3D interaction change. Conversely, there is the possibility that explicitly modeling ChIP enrichment may decrease the power to detect interactions or differential interactions by normalizing away the true signal. In our analyses, we found evidence that the latter is the case. We first found that normalizing for ChIP-seq bias by including this signal either as a continuous or categorical covariate in the HiC-DC+ model decreased power to detect CRISPRi-FlowFISH validated interactions (Supplementary Note 1 and Supplementary Fig. 8). In fact, since HiC-DC+ covariates such as GC content and mappability account for sources of systematic biases in high-throughput sequencing, we may already be implicitly correcting for a component of ChIP bias. Second, we systematically examined differential HiC-DC+ H3K27ac HiChIP results between two ENCODE cell lines using parallel Hi-C and H3K27ac ChIP-seq data (Supplementary Note 3). This analysis revealed that HiC-DC+ differential HiChIP events generally do not occur due to ChIP bias alone. Rather, we found that most differential HiChIP interactions with a concordant differential ChIP signal at one or both anchors exhibit simultaneous and concordant differential Hi-C signal. Promoter-anchored differential HiChIP interactions with no Hi-C signal change but differential ChIP at the promoter-distal anchor were enriched for differential target gene expression and depleted for significant Hi-C interactions. This suggests enrichment for true regulatory interactions that are not detected in Hi-C due to lack of sensitivity. We acknowledge that these differential analyses were limited to H3K27ac HiChIP and ChIP-seq due to availability of data sets. Our results confound the notion of normalization by ChIP in the case of regulatory interactions and address an important gap in the literature on this topic.

Several of our analysis vignettes demonstrated the effectiveness of HiC-DC+ for detecting significant and differential regulatory interactions, both in H3K27ac HiChIP experiments and in very high-coverage Hi-C data sets. In particular, using deep Hi-C data with ~5B reads per condition in the THP-1 cell line model of monocyte to macrophage differentiation, we found widespread gain of putative enhancer–promoter interactions in the macrophage state that were associated with gain of accessibility and H3K27ac at the non-promoter anchor and upregulation of targeted genes. By contrast, the widely used HiCCUPS method is well suited for detection of CTCF-mediated structural loops but

appears to lack sensitivity for regulatory loops. Differential analysis applied to HiCCUPS loops identified few differential interactions, and those largely had smaller effect size (log fold change), showing the dichotomy between the relative stability of the structural interactome and the dynamics of the regulatory interactome. Therefore, as a statistically principled and all-purpose significant and differential interaction caller, HiC-DC+ complements structural loop callers and empowers examination of the role of 3D genomic interactions in the regulation of gene expression.

## Methods

**Pre-processing of Hi-C and HiChIP data**. We aligned HiChIP reads to hg19 or mm10 genomes and filtered out reads that are duplicate or invalid ligation products using the HiC-Pro pipeline (v_2.11.1) with default settings. We processed Hi-C reads using Juicer pipeline (v_1.7.6) with default settings. For benchmarking with CRISPRi in mESCs from Fulco et al.[11], we used mESC H3K27ac HiChIP reads aligned to mm9. For all the other mouse data, we mapped reads to mm10 genome.

**TAD annotations**. We used sub-TAD annotations from Arrowhead for GM12878 Hi-C data (GSE63525). We found TADs at 50 kb resolution using TopDom (v_0.0.2) with "w" as 10 on ICE normalized Hi-C counts.

**Calling loops using FitHiChIP, MAPS, and hichipper**. We used 5 kb-binned hichipper interactions in GM12878 SMC1A and K562 H3K27ac HiChIP reported by Bhattacharyya et al., 2019[21] (GM12878 SMC1A from Table_GC-ALL table and K562 H3K27ac from Table_KH-ALL table). MAPS and FitHiChIP (L) interactions (FDR <0.01) were also obtained from Bhattacharyya et al., 2019[21] (GM12878 SMC1A from Table_GC-ALL table) and interactions, called using pooled replicates, up to 1.5M were used for comparisons.

We ran MAPS (v_1.1.0) for H3K27ac HiChIP in K562 data with HiChIP inferred ChIP peaks (Table Table_KH-L-H from[21], with the following parameters: "bin_size = 5000; fdr = 0; filter_file = None; generate_hic = 0; mapq = 30; length_cutoff = 1000; threads = 10; per_chr = True, binning_range = 1500000" using pooled allValidPairs file.

We ran FitHiChIP (v_8.1.0) on pooled H3K27ac HiChIP in K562 data with HiChIP inferred ChIP peaks (Table Table_KH-L-H from[21]), with the following parameters: IntType = 3; BINSIZE = 5000; LowDistThr = 5000; UppDistThr = 1500000; UseP2PBackgrnd = 0; BiasType=1; MergeInt=1; QVALUE = 0.01" by providing pooled allValidPairs file.

**Benchmarking with CRISPRi-FlowFISH results**. We use the 5091 candidate enhancer–promoter-target gene data from Fulco et al.[11] reported in Supplementary Table 6a for K562 cells and add 15 singleton experiments for candidate enhancer–promoter pairs along with their tested target genes[28] for K562 cells reported in Table S3A. Because other methods (all except ABC and HiC-DC+) do not check for interactions shorter than 5 kb (17 candidates) and HiC-DC+ longer than 1.5 Mb (324 candidates) for K562 H3K27ac HiChIP, we removed these candidates from consideration. Among those within the distance range, we also removed candidate elements on gene promoters (1147 candidates) following Fulco et al.[11]. This leaves a set of 3618 candidate enhancer–promoter-target gene experiments. We treated enhancer–promoter pairs reported at FDR < 0.05 as significant enhancer–promoter candidates for the tested target gene.

We also use validated enhancer–promoter pairs in mESCs from Fulco et al.[11], reported in Table S6b. There are a total of 57 regulatory element-gene pairs (excluding promoter–promoter pairs) for a total of 56 genes. Fourteen of those elements were found to significantly regulate target gene expression using CRISPRi, while the rest were not. We performed our benchmarking on these 57 enhancer–promoter pairs.

**Effect size comparison**. We collected significant (FDR < 0.01) looping events identified in K562 H3K27ac HiChIP at 5 kb bins overlapping with candidate enhancer–promoter pairs for each of the benchmarked methods (PET count ≥2 for hichipper) and compared the reported fraction change in gene expression using one-sided Wilcoxon rank sum test with the alternative hypothesis being HiC-DC+ having greater reduction in gene expression on the CRISPR-perturbed candidates.

**Gene-wise performance comparison**. We grouped 3618 candidate enhancer–promoter pairs by their 71 target genes, chose 22 target genes (2609 candidates) that has at least one significant and one insignificant pair, and computed the auPR for each target gene and for each benchmarked method using significance scores collected from overlapping 5 kb bins for each candidate. We ordered the precision-recall curve in an increasing order of $P$ values for HiC-DC+, FitHiChIP, and MAPS, and in decreasing order of PET counts for hichipper. We compared the range of auPR values across methods using one-sided Wilcoxon signed rank tests with the alternative hypothesis being HiC-DC+ having greater auPR. For the

comparison between ABD and ABC methods (Wilcoxon signed rank test, $P = 0.39$, $N = 23$, alternative: ABD is greater), we included candidate enhancer–promoter pairs closer than 5 kb as well.

**Finding CTCF motif orientation**. We found the CTCF motif orientation of the significant GM12878 SMC1A HiChIP loop anchors (FDR <0.01, 20 kb ≤ loop distance ≤ 1.5M) identified by each method using Juicer tool command Motif-Finder (v_1.7.6). We used IDR-thresholded GM12878 CTCF peaks available from ENCODE (ENCFF710VEH).

**SubTAD metaplot generation**. For each method, we found the number of significant GM12878 SMC1A HiChIP interactions (FDR < 0.01, 20 kb ≤ loop distance ≤ 1.5M) a 5 kb bin makes with other regions. Then, using deepTools (v_3.1.1) computeMatrix "scale-regions" and plotProfile, we generated a metaplot of significant interactions over GM12878 subTADs (GSE63525).

**APA plot generation**. We used Juicer Tools (v1.6.2) to perform APA on the corresponding Hi-C data with all default settings either at a resolution of 5 or 25 kb (for comparison of differential calling methods on HAP1 data). APA output were then $z$-score normalized to retrieve pixel intensities, and then rescaled to $[-1,1]$. Smoothed images were then rendered using ggplot. We used the corresponding Hi-C data to generate APA plots for HiChIP calls. For each APA plot, P2LL scores were computed to determine enrichment of the signal to the lower left corner and reported as APA scores.

**Statistical modeling of interaction bin count data**. HiC-DC+ follows the feature selections and model specifications outlined in Carty et al.[9] with the choice of a NB model for the read counts rather than a zero-truncated negative binomial distribution. Specifically, we used a GLM based on NB regression to model the Hi-C read counts, and we used the fitted model to estimate the statistical significance ($P$ value) to the Hi-C interaction bin counts. Following previous notation[9], we let $\mathbf{Y} = (y_{ij})$ represent the Hi-C contact map of intra-chromosomal interactions, where $i$ and $j$ are a pair of genomic intervals (either through uniform binning of the genome or fixed size bins of consecutive RE fragments) and the tuple $(i,j)$ defines an interaction bin. Each bin has an associated vector of covariates, which we denote as $\mathbf{X} = (x^{\text{dist}}; x^{\text{gc}}; x^{\text{map}})$, and $y_{ij}$ is modeled as a random variable that follows a NB distribution. The regression model is defined as: $P(\mathbf{Y} = y_{ij}|\mathbf{X}) = f(k;\mu_{ij}; \alpha)$, where the distribution $f(k;\mu_{ij}; \alpha)$ is a NB distribution with dispersion parameter $a$. The NB mean parameter $\mu_{ij}$ is described with a log-linear model:

$$\log(\mu_{ij}) = \beta_0 + \sum_k \beta_k B_{k,l}(x_{ij}^{\text{dist}}) + \beta_{\text{gc}} x_{ij}^{\text{gc}} + \beta_{\text{map}} x_{ij}^{\text{map}} \qquad (1)$$

where $\beta_0$ is the intercept term, and $\beta_{\text{gc}}$ and $\beta_{\text{map}}$ are coefficients for GC content and mappability features, respectively. In the case of uniform binning, the effective bin size due to RE site occurrences is included as a covariate, as in the HiC-DC model. We modeled the relationship between genomic distance and contact significance with a third order B-spline, which takes the distance covariate as input, and the B-spline basis functions are as previously described[9]. Specifically, we use six degrees of freedom for fitting the spline and have three inner knots. These inner knots are at the 25, 50, and 75% quantiles of the linear genomic distance.

To increase the statistical power of the model, we removed bins with counts that exceed the 97.5% percentile of null distribution that we deem as positive outliers, corresponding to potentially nonrandom contacts, and then refit the model to the remainder of the data. We use the function glm.nb in the R library *MASS* to fit the NB regression model.

HiC-DC+ assesses the significance of bin counts based on the corresponding estimated NB distribution for that bin. The $P$ value associated with each interaction bin becomes 1 minus the cumulative distribution function fit for that bin given its feature values, i.e., $1 - \sum_{k=0}^{y_{ij}} f(k;\mu_{ij}; \alpha)$. Significant bins can then be selected on the basis of adjusted $P$ values following the Benjamini–Hochberg procedure to control for FDR.

The R package *HiCDCPlus* allows for model selection for other interaction bin count data by providing an option to choose the zero-truncated NB distribution, as well as optional parameters to (i) add/remove/change local genomic features as desired and (ii) use the logarithm of genomic distance as the distance covariate instead of using B-splines.

**Differential interaction calling**. We call differential interactions using DESeq2 and raw counts, with two modifications. First, analyzed loci are the interaction bins in the union set of significant interactions (FDR < 0.1) across all conditions to be pairwise tested. Second, size factors for these loci are determined for each genomic distance separately. Following DESeq2 assumptions, we enforce that the median normalized count for interaction bins at each genomic distance should be similar across conditions. Specifically, for a given distance $|i - j| = d$, the size factor for condition $k$ is chosen to be $\widehat{s}_d = \text{median}_{|i-j|=d} y_{ij}^k$, so that normalized counts become $\frac{y_{ij}^k}{s_d}$.

**Calling differential interactions using other tools**. For H3K27ac HiChIP data in mESC and MEF[13], we called differential loops at 5 kb using diffloop (v_1.16.0)[27] using R version 3.6. Since the published interactions were called by an approach similar to Mango[29], we implemented the Mango correction function to remove biases associated with the method. The loops were then filtered at FDR < 0.05 and absolute fold change >1.5. We performed the differential contact calling on Hi-C data in HAP1 and WAPL knockout HAP1 cells at 25 kb resolution using mutliHiCcompare[25], Selfish[26], and diffHic[24]. All differential interactions were filtered at FDR < 0.05 and absolute fold change >1.5. After loading raw matrices into multiHiCcompare (v_1.6.0), we normalized samples with cyclic loess normalization and called the differential interactions by exact test. The python3 version of Selfish (v_1.10.2) was used to detect differential interactions by applying the method to HiC-DC+ (O/E) normalized matrices. For diffHic (v_1.14.0) analysis, we loaded raw contact count data as InteractionSet objects, filtered by average counts, used loess normalization, and applied the tool's statistical model function to estimate $P$ values.

**ChIP-seq analysis**. We aligned CTCF and H3K27ac ChIP-seq reads to hg19 using BWA (v_0.7.17-r1188). Then, we extracted uniquely aligned paired reads using SAMtools (v_1.9) and removed PCR duplicates using Picard tools (v_2.18.16). Peak calling was performed for each individual and pooled replicates of each cell type using MACS2 (v_2.1.2) with parameters "-g hs -p 0.0". To find reproducible peaks across replicates for each histone mark, we calculated the irreproducible discovery rate (IDR) using IDR (v_2.0.3) with parameters "—samples rep1.narrowPeak rep2.narrowPeak—peak-list pooled.narrowPeak -o—plot". We combined peaks passing an IDR threshold of 0.05 in each condition for each mark. We used featureCounts (v_1.6.4) to obtain ChIP-seq read counts in the peak atlas, and applied DESeq2 (v_1.24.0) to these counts to find the differential occupancy of each mark between conditions. Bedtools genomeCoverageBed (v_2.27.1) was used to generate bedgraph files scaled with DESeq2 sample size factors and bedgraph files were converted to bigwig using UCSC bedgraph2bigwig (v_4).

**ATAC-seq analysis**. We trimmed the raw reads and filtered for quality using cutadapt (v_2.3). Trimmed reads were aligned to the hg19 genome using Bowtie2 (v_2.3.4.3) and uniquely mapped reads were retained. After centering the reads on the transposase binding event by shifting all positive-strand reads by 4 bp downstream and all negative-strand reads 5 bp upstream, peak calling was performed on each replicate and all replicated pooled together using MACS2 (v_2.1.2) with parameters "-g hs—nomodel". We calculated the IDR using IDR (v_2.0.3) to find reproducible peaks across replicates. We combined peaks passing an IDR threshold of 0.05 to generate an atlas of peaks. Differential accessibility analysis of the peaks and generation of normalized bigwig tracks was performed as described in ChIP-seq analysis.

**Annotation of peaks**. We used the GENCODE transcript annotations of the hg19 (v19) genome to define the genomic location of transcription units. ATAC-seq and ChIP-seq peaks were annotated as promoter peaks if they were within 2 kb of a transcription start site. Non-promoter peaks were annotated according to the relevant transcript annotation in the following order of priority: intronic, exonic, or intergenic. Intergenic peaks were assigned to the gene whose TSS or 3′ end was closest to the peak. In this way, each peak was unambiguously assigned to one gene.

**RNA-seq and analysis**. We conducted differential gene expression analysis on the gene counts tables provided in GSE96800 using DESeq2 (v_1.24.0).

**Comparison of macrophage to monocyte differential loops**. We reported shared looping events by calling HiC-DC+ differential interactions at 10 kb at an FDR of 0.01 and overlapping with interactions reported by Phanstiel et al.[14], requiring anchors for our interactions to fall within a 10 kb flank of previously the reported anchors.

**Candidate enhancers and promoter-anchored loops**. To identify candidate regulatory elements, we first assembled the atlas of reproducible H3K27ac ChIP-seq and ATAC-seq peaks in each the two conditions filtered at an IDR threshold of 0.05. We defined candidate enhancers by taking the intersection of genomic intervals containing both an H3K27ac ChIP and an ATAC peak from this atlas. We identified promoter regions as within 2 kb of transcription start sites (TSS) of genes annotated by GENCODE GRCh37.p13. We identified elements with differential accessibility or activity of candidate enhancers as based on DESeq2 applied to ATAC or H3K27ac read counts, respectively, with adjusted $P < 0.05$. We described candidate enhancers as static if neither signal was significantly differential. We overlapped CTCF peaks called at IDR 0.05 with candidate enhancer regions and similarly defined differential and static CTCF binding events.

We define promoter-anchored loops as loops and those where one of the anchors overlaps a promoter region. Likewise, enhancer–promoter interactions are promoter-anchored loops with one anchor overlapping a promoter region and the other anchor overlapping a candidate enhancer from the atlas.

**GREAT analysis of differential looping events**. We used enhancer–promoter interactions as defined above for GREAT (v_4.0.4) analysis. We filtered results for significance (hypergeometric test, adjusted $P < 0.05$) while restricting to GO terms associated with at most 300 genes.

**Reporting summary**. Further information on research design is available in the Nature Research Reporting Summary linked to this article.

## Data availability

All data sets used in this study are publicly available and summarized in Supplementary Data 1 with accession codes. All relevant analyzed data is available upon request.

## Code availability

The source code and the documentation for the HiCDCPlus R package is available at https://bitbucket.org/leslielab/hicdcplus.

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

## Acknowledgements

This work was supported by NIH/NHGRI U01 award HG009395 and NIH/NIDDK U01 award DK128852.

## Author contributions

M.S. and C.S.L. conceived of the project, designed the statistical methods, and wrote the manuscript. M.S. developed and implemented the methods, developed the software package, and performed benchmarking analyses. W.W. performed computational analyses for monocyte to macrophage differentiation and contributed to writing the manuscript. Y.Z. performed benchmarking experiments for differential analysis under the supervision of R.K. K.V.D. performed statistical analyses for HiChIP benchmarking. C.S.L. supervised the research. All authors read and approved the final version of the manuscript.

## Competing interests

The authors declare no competing interests.
