## [Peer Review File · Nature Communications]

Reviewers' Comments:

Reviewer #1:

Remarks to the Author:

In this paper, the authors proposed HiC-DC+, a new method to identify significant chromatin interactions from Hi-C and HiChIP datasets, and perform differential analysis. This work addressed an important research question. The authors benchmarked the performance of HiC-DC+ with other existing methods in multiple datasets. The software website in Bitbucket is well maintained with detailed user manual. The paper is also well written. However, I am not convinced that HiC-DC+ achieved substantial improvement over existing methods. The authors need to perform additional analysis to better demonstrate the advantage of HiC-DC+ in practice. Here are my specific comments.

Major comments:

1. HiC-DC+ is an extension of their previous work HiC-DC (PMID: 28513628). It would be helpful to discuss the key differences between HiC-DC and HiC-DC+, and add HiC-DC into the comparison. In addition, FitHiChIP has two modes: peak-to-peak or stringent (S) and peak-to-all or loose (L), with extra merging step (M) to improve the specificity. Since the original FitHiChIP paper (PMID: 31530818) evaluated all 4 combinations (S, S+M, L, L+M), I hope the authors can also compare these 4 combinations of FitHiChIP with HiC-DC+.
2. The authors used CRISPRi-FlowFiSH data for 22 genes in K562 cells (Fulco et al, 2019) as the gold standard, which is a limited set. In addition, they can also use HiCCUPS loops identified from deeply sequenced Hi-C data from K562 cells (Rao et al, 2014) as the gold standard. For example, for the same number of top interactions (top 1000, top 2000, top 5000, top 10,000, etc) identified by each method, how many HiCCUPS loops can be recovered?
3. Fulco et al 2019 paper also contains the validated enhancer-promoter pairs in mESCs (Table S6b). Many mESC Hi-C and HiChIP datasets are also publicly available (PMID: 29053968, PMID: 27643841). I hope the authors can also evaluate the performance of different methods using mESC data.
4. In Fig 2a, the improvement of HiC-DC+ over FitHiChIP looks incremental. In addition to auPR, the authors need to show both precision and recall for each method.
5. When applying HiC-DC+ to HiChIP data, how did the authors normalize the bias from different ChIP enrichment levels? The HiC-DC+ identified interactions may be biased towards loci with stronger ChIP enrichment level. In addition, when performing differential analysis between HiChIP data from two different samples, the sample-specific ChIP enrichment level can also introduce bias to the identified differential interactions. The authors need to specifically model ChIP enrichment bias in both significant interaction detection and differential interaction analysis.
6. What is the reproducibility of HiC-DC+ for detecting significant interactions and differential interactions?
7. Whether the HiC-DC+ results are consistent at different bin resolutions (for example, 25Kb, 10Kb and 5Kb)? Can it be applied to nucleosome resolution Micro-C data (PMID: 32213324)?
8. When applying HiC-DC+ differential analysis to biological replicates of the same cell types, for examples, replicates of mESC Hi-C data (GSE96107) or replicates of mESC HiChIP data (GSE80820), all the identified differential interactions will be false positives. Can HiC-DC+ control type-I error in this scenario? Since HiC-DC+ identified much more differential interactions than other methods, I am concerned that HiC-DC+ results may contain high percentage of false positives.

Minor comments:

1. In Fig S3, the red curve (HiC-DC+) and blue curve (FitHiChIP) look almost identical. I am not convinced that HiC-DC+ shows higher enrichment at TAD boundaries than FitHiChIP.
2. The differential analysis of HAP1 data is at 25Kb bin resolution (Fig 3). To better measure enhancer-promoter interactions, I hope they can perform such differential analysis at 5Kb and 10Kb bin resolution.
3. The authors can provide more details on how they handle insertions, deletions and copy number variations in the cancer cell line K562. Those structural variations can introduce false positives in the identified significant interactions.
4. As a new computational method, the authors need to provide more details on the computational cost (such as memory and running time), and compare that with other existing methods.

Reviewer #2:

Remarks to the Author:

Sahin et al. created an update to HiC-DC called HiC-DC+. They evaluated the effectiveness of significant interaction identification in HiChIP data and the differential analysis of interactions from Hi-C and HiChIP experiments. The effort to integrate HiChIP interaction identification and statistical differential interaction calling is worthwhile and there are several packages that try to accomplish similar tasks that have been implemented since the creation of the original HiC-DC algorithm.

1. It was difficult to tell whether the program has significantly improved or not. The paper would be dramatically improved by a better description of why this update represents a significant advance in the field. From what I can tell, most of the analysis could have been done with the original HiC-DC algorithm in conjunction with DESeq2. The way the manuscript is written, this update seems to simply improve storage and parallelization and does not represent a departure from current methods. The key advances over the original algorithm should be made clearer.
2. Using the CRISPRi-FlowFISH data to evaluate H3K27ac HiChIP interaction calling is a clever idea. However, HiChIP may display interactions that do not appear impactful in the CRISPRi-FlowFISH. This is because CRISPRi-FlowFISH is used to describe the impact that an enhancer has on gene expression. Interaction signal is only one piece of that puzzle as noted by the ABC model that the authors have cited. Thus, simply because CRISPRi-FlowFISH does not detect an impact on the gene's expression, does not mean that there was not an interaction.
3. In addition to CRISPRi-FlowFISH, the authors should use other methods to evaluate their significant HiChIP interactions. For example, they should include APA plots (average metaplots) and APA scores for interactions unique to HiC-DC+, those shared between algorithms, and those unique to other algorithms.
4. Differential HiChIP and differential Hi-C interactions should be evaluated in a similar fashion (APA plots in each sample and for unique vs shared differential interactions). This will allow the reader to evaluate how well each algorithm describes observable differences between maps.
5. Line 147-148: "only Hi-C-DC+ recovered both findings: a majority of gained interactions connected neighboring TADs together with a drastic loss of intra-TAD interactions.". It is unclear why the authors make this claim. Each of the other methods seem to show a similar effect.
6. I am very concerned that HiC-DC+ does not incorporate self-ligation / IP peak detection during HiChIP identification, particularly when thinking about differential interaction identification. Because HiChIP involves immunoprecipitation of interactions, any signal differences between samples can either be due to loss of interactions, or simply due to loss of immunoprecipitation efficiency. Thus, while the actual interactions within the cell may be unchanged, the HiChIP signal can be changed simply due to changes in the protein binding. Incorporating immunoprecipitation signal (ChIP-seq or self-ligations) is an important step to delineate interaction changes vs signal changes that are more ambiguous. It would be best to report both types of changes as distinct categories, but that requires

estimating the IP efficiency of each locus. Without that step, the differential interaction calling is ambiguous.

7. The authors should explain why they see the ABD “outperforms the ABC score” (Supplementary Note). The cited ABC paper already performed a comparison to a model with distance (ABD) instead of contact (ABC) and found that the ABC model was better.

8. The authors compare to differential loop calling in Phanstiel et al, which used HiCCUPS. This program was designed to identify punctate CTCF loops. Others have designed loop calling algorithms for a similar purpose (SIP, Rowley et al., Genome Res 2020 and Mustache, Ardakany et al., Genome Biology 2020). These only identified ~13,000, or ~18,000 loops in human cells. Yet the authors claim to have 64,844 differential loops. The types of interactions (CTCF loops vs enhancer-promoter interactions) that are identified by HiC-DC+ is therefore probably different from CTCF loop callers like HiCCUPS. If the intention is to call enhancer-promoter interactions, it should be clearly stated that the intention of these programs is quite different. Additionally, these HiC-DC+ interactions should not be contrasted with HiCCUPS or other callers where the intent is different.

9. The authors should provide the number of total loops called by each method for each dataset, including each Hi-C and Hi-ChIP datasets. This is important to evaluate differences between algorithms.

10. It is known that sequencing depth impacts Hi-C data analysis, particularly regarding feature identification and differential analysis. There is a trade-off between using replicates separately (as DESeq2 incorporation requires) vs pooling replicates to obtain deeper matrices. It would be valuable for the authors to evaluate how sequencing depth affects HiChIP feature calling, and how it may affect differential interaction calling. On a side note, it would also be valuable to evaluate how imbalanced depth between two replicates could impact differential interaction calling.

11. I didn't see any of the usual benchmarking. How much memory does HiC-DC+ use for interaction calling in Hi-C, HiChIP, and differential interaction calling? How long does it take with various computing power?

Response to the Reviewers' Comments

We thank the reviewers for their comments and critique to help further strengthen our manuscript. We have revised the paper by including new analyses, expanding on the text to delineate our arguments better, and providing a detailed explanation below.

Reviewer #1 (Expertise: Chromatin organization using HiC or HiChIP):

In this paper, the authors proposed HiC-DC+, a new method to identify significant chromatin interactions from Hi-C and HiChIP datasets, and perform differential analysis. This work addressed an important research question. The authors benchmarked the performance of HiC-DC+ with other existing methods in multiple datasets. The software website in Bitbucket is well maintained with detailed user manual. The paper is also well written. However, I am not convinced that HiC-DC+ achieved substantial improvement over existing methods. The authors need to perform additional analysis to better demonstrate the advantage of HiC-DC+ in practice. Here are my specific comments.

Major comments:

1. HiC-DC+ is an extension of their previous work HiC-DC (PMID: 28513628). It would be helpful to discuss the key differences between HiC-DC and HiC-DC+, and add HiC-DC into the comparison. In addition, FitHiChIP has two modes: peak-to-peak or stringent (S) and peak-to-all or loose (L), with extra merging step (M) to improve the specificity. Since the original FitHiChIP paper (PMID: 31530818) evaluated all 4 combinations (S, S+M, L, L+M), I hope the authors can also compare these 4 combinations of FitHiChIP with HiC-DC+.

We ran FitHiChIP on K562 H3K27ac HiChIP for the three other modes (S, S+M, and L+M) as well. The best performing among the four is the FitHiChIP (L), which is one that we used for the CRISPRi-FlowFISH method comparison on the main text. We now present these additional results in **Supplementary Fig. 4**.

HiC-DC uses hurdle negative binomial regression while HiC-DC+ utilizes negative binomial regression. We also ran HiC-DC on K562 H3K27ac HiChIP to compare both methods and HiC-DC+ outperformed the previous version in per gene auPR values as validated by CRISPRi-FlowFISH (**Fig. R1** below, P value < 0.01, Wilcoxon signed-rank test). ~80% of the significant interactions (FDR < 0.01) identified with HiC-DC+ are also found to be significant by HiC-DC; however, HiC-DC+ captures more interactions shorter than 500kb. The major difference between two tools lies in the efficiency of the code, integrability with common pre-processed data formats, and added functionality including but not limited to detection of differential interactions and TAD annotation. The efficiency of the tool is now summarized in **Supplementary Tables 1-5**. Running the HiC-DC model with 8 cores in parallel on chromosome 1, 11 and 22 takes 13, 7, and 3 min and requires 14.2, 9, and 5 GB of memory, respectively; whereas running the HiC-DC+ model with 8 cores in parallel for the entire genome for the same data only takes 3 min and requires 17.1 GB of memory at the same resolution.

Figure R1. Evaluation of method performance by per gene auPR values. For each target gene, we ranked the candidate regulatory elements tested by CRISPRi-FlowFISH in K562 cells based on the significance (P value) of the HiChIP interactions with the promoter as estimated by each method.

2. The authors used CRISPRi-FlowFiSH data for 22 genes in K562 cells (Fulco et al, 2019) as the gold standard, which is a limited set. In addition, they can also use HiCCUPS loops identified from deeply sequenced Hi-C data from K562 cells (Rao et al, 2014) as the gold standard. For example, for the same number of top interactions (top 1000, top 2000, top 5000, top 10,000, etc) identified by each method, how many HiCCUPS loops can be recovered?

HiCCUPS is quite conservative and its interaction calls are not enriched for enhancer-promoter interactions, as we show in our analysis of THP-1 monocyte to macrophage differentiation (see also comment 8 from Reviewer #2 that characterizes HiCCUPS as a CTCF loop caller and not designed for detection of enhancer-promoter interactions). That is why we believe it is not a good standard for H3K27ac HiChIP data. However, we performed the benchmarking as requested using the same number of top interactions identified by HiC-DC+ and FitHiChIP(L) and overlapping with HiCCUPS loops from Rao et al., 2014 for K562 Hi-C (total of 5867 interactions for $0 < D \leq 1.5M$). The overlap is small as expected, but HiC-DC+ recovers considerably more loops than FitHiChIP(L) (**Table R1** below).

Table R1. Comparison of methods based on the overlap with HiCCUPS loops.

Top cut-off	HiC-DC+	FitHiChIP(L)
1000	41	24
2000	74	47
5000	149	110
10000	240	181

3. Fulco et al 2019 paper also contains the validated enhancer-promoter pairs in mESCs (Table S6b). Many mESC Hi-C and HiChIP datasets are also publicly available (PMID: 29053968, PMID: 27643841). I hope the authors can also evaluate the performance of different methods using mESC data.

We thank the reviewer for this suggestion. We analyzed mESC H3K27ac HiChIP data from Mumbach et al., 2017 and called interactions using HiC-DC+ and FitHiChIP(L and L+M). There were a total of 57 regulatory element-gene pairs (excluding promoter-promoter pairs) for a total of 56 genes in Table S6b in Fulco et al., 2019. 14 of those elements were found to significantly regulate target gene expression using CRISPRi, while the rest were not.

We compared the performance of HiC-DC+ and FitHiChIP(L and L+M) by the overall auPR (**Table R2** below). HiC-DC+ outperformed FitHiChIP for identifying the enhancer-promoter interactions from mESC H3K27ac HiChIP. We also found that HiC-DC+ was more successful in identifying promoter-enhancer loops of enhancers with large effect size on target gene expression (**Supplementary Fig. 6**). We ran FitHiChIP using two different mESC H3K27ac ChIP-seq peak lists from ENCODE (ENCSR000CGQ): one is replicated (n=25,928) and the other is called from one replicate only (n=142,117). When non-replicated peaks were used, FitHiChIP(L) called 3 times more significant interactions. Only 1 of the significant (FDR<0.01) FitHiChIP(L+M, non-replicated peaks) interactions overlapped with the 57 candidate pairs, while none of the significant (FDR<0.01) FitHiChIP(L+M, replicated peaks) interactions had an overlap. We have left out hicchipper and MAPS, as we already substantially outperformed them in other comparisons. These new results are now included as **Supplementary Fig. 6**.

Table R2. Evaluation of method performance by overall auPR values on mESC CRISPRi data.

Method	auPR
HiC-DC+	0.66
FitHiChIP (L)	0.20
FitHiChIP (L+M)	0.29
ABC	0.73
ABD	0.80

4. In Fig 2a, the improvement of HiC-DC+ over FitHiChIP looks incremental. In addition to auPR, the authors need to show both precision and recall for each method.

We report precision and recall for interactions called significant at $FDR < 0.05$ and < 0.01 for HiC-DC+ and FitHiChIP in **Table R3** on the data used to generate **Fig 2a**. We also provide the full precision-recall curve for this data now in the supplementary figures as **Supplementary Fig. 5**. HiC-DC+ outperforms all methods at precision up to 20% recall and performs similarly to FitHiChIP at higher recall values (and lower precision values, **Supplementary Fig. 5**).

Table R3. Comparison of method performance by precision and recall for interactions called significant at $FDR < 0.05$ and < 0.01 for each method.

Method	FDR	Precision	Recall	Overall auPR
HiC-DC+	0.05	0.092	0.623	0.205
FitHiChIP (L)	0.05	0.063	0.846	0.180
HiC-DC+	0.01	0.107	0.576	
FitHiChIP (L)	0.01	0.067	0.846	

5. When applying HiC-DC+ to HiChIP data, how did the authors normalize the bias from different ChIP enrichment levels? The HiC-DC+ identified interactions may be biased towards loci with stronger ChIP enrichment level. In addition, when performing differential analysis between HiChIP data from two different samples, the sample-specific ChIP enrichment level can also introduce bias to the identified differential interactions. The authors need to specifically model ChIP enrichment bias in both significant interaction detection and differential interaction analysis.

We believe that the issue of whether to normalize for ChIP-seq signal in HiChIP analyses is not adequately addressed in the literature. On one hand, there is the concern raised by the reviewer

that without normalization, HiChIP interactions may be biased for higher ChIP enrichment, and differential HiChIP interactions may be due to differential ChIP enrichment rather than an underlying 3D interaction change. On the other hand, there is the possibility that explicitly modeling ChIP enrichment may *decrease* the power to detect interactions or differential interactions by normalizing away the true signal. In the results we present below, we argue that the latter is the case. First, we show that normalizing for ChIP-seq signal in our model decreases power to detect CRISPRi-FlowFISH validated interactions. Second, we compare our differential HiChIP results systematically to differential Hi-C signal to show that the anticipated problem -- differential HiChIP interactions due only to differential ChIP without underlying differential Hi-C signal -- is not a major issue.

First we modelled ChIP enrichment explicitly by adding any of the following covariates to the model for K562 H3K27ac HiChIP data (**Supplementary Note 1**):

1. chip: standardized log transformed H3K27ac ChIP intensity
2. cov_counts: interaction bin counts divided by the average count over bins that share the same peak status (i.e., both anchors have a peak (peak-to-peak), one has a peak (peak-to-all), neither anchor has peak (all-to-all)) similar to FitHiChIP
3. peak: peak status as described above as a categorical covariate with three levels
4. cov: standardized log transformed ChIP enrichment level measured by the number of short-range reads (intra-chromosomal reads \leq 1kb), similar to MAPS

Then, we tested the performance of these models on CRISPRi-FlowFISH data by per gene auPR values and included this analysis as **Supplementary Fig. 8**, which we mention in **Results** and **Discussion** as well.

Supplementary Fig. 8 and **Table R4** (below) show that including ChIP enrichment as a covariate worsens the performance for enhancer-promoter interactions from CRISPRi-FlowFISH data ($P < 0.05$, Wilcoxon signed rank test). These models also exhibit a reduced overlap with K562 HiCCUPS loops (**Table R5** below).

All these observations suggest that explicitly modeling ChIP enrichment might actually *decrease* power to detect specific events, thus we choose not to include it in our model.

Table R4. Comparison of method performance by precision and recall for interactions called significant at FDR < 0.05 and <0.01 for each method.

Method	FDR	Precision	Recall	Overall auPR
HiC-DC+	0.05	0.092	0.623	0.205
HiC-DC+ (chip)	0.05	0.047	0.008	0.054
HiC-DC+ (cov_counts)	0.05	0.044	0.369	0.124
HiC-DC+	0.01	0.107	0.576	
HiC-DC+ (chip)*	0.01	0.000	0.000	
HiC-DC+ (cov_counts)	0.01	0.045	0.338	

*: No true positive peaks identified with HiC-DC+ (chip) at 1% FDR

Table R5. Comparison of methods based on the overlap with HiCCUPS loops.

Top cut-off	HiC-DC+	HiC-DC+ (chip)	HiC-DC+ (cov_counts)
1000	41	27	15
2000	74	56	26
5000	149	126	50
10000	240	245	81

To see whether our differential HiChIP interactions are mostly driven by ChIP enrichment bias or the underlying chromatin architecture changes, we found differential H3K27ac HiChIP interactions (FDR < 0.05 & |logFC| >1) between K562 and GM12878 (Mumbach et al., 2017, Rao et al., 2014) as well as differential H3K27ac ChIP-seq peaks (FDR < 0.05) (**Supplementary Note 3**, also added a new subsection in the **Results**). We found that gained/lost HiChIP interactions are closely associated with increased/decreased Hi-C signal as well as increased/decreased ChIP signal in one of the anchors (now **Supplementary Fig. 21**). This suggests that differential HiChIP interactions are driven not only by ChIP signal but also Hi-C. To further support our claim, we grouped differential HiChIP interactions based on whether none, one, or both anchors overlap with differential ChIP-seq peaks such that:

- chip_sig_sig: Both anchors have an overlapping differential ChIP-seq peak

- chip_sig_stable: One of the anchors have an overlapping differential ChIP-seq peak, the other anchor does not,
- chip_stable: None of the anchors have such an overlapping differential ChIP-seq peak.

We further grouped each of these three groups of interactions into whether the ChIP-seq peaks overlapping with the associated anchors had exhibited a positive log-fold change (“up”) or a negative log-fold change (“down”) on average.

Supplementary Fig. 27a shows the change in normalized Hi-C (O/E) counts for each of these 6 categories in lost H3K27ac HiChIP interactions in K562 over GM12878, whereas **Supplementary Fig. 27b** shows the respective change in gained H3K27ac HiChIP interactions in K562 over GM12878. Note, as expected, that the sign of the average ChIP-seq signal change in the anchors (blue vs. red bars in **Supplementary Fig. 27a,b**) largely coincided with the sign of the HiChIP interaction (e.g. almost all chip_sig_sig interactions that are lost in K562 vs GM12878 also lose ChIP-seq signal). Importantly, most of our differential HiChIP interactions exhibit change in Hi-C interactions in the same direction independent of the ChIP status. Indeed, the differential HiChIP interactions with strongest concordant ChIP signal changes (chip_sig_sig) had the *largest* differential Hi-C signal as well. Rather than HiChIP finding differential interactions solely due to ChIP signal, we found that the majority (61%) of differential HiChIP interactions with differential ChIP-seq signal at one or more anchors were supported by concordant differential Hi-C signal ($|\log_2FC \text{ of normalized counts}| \geq 0.58$; i.e., $FC \geq 1.5$ or $FC \leq 0.67$).

Conversely, we also found some (27%) differential HiChIP interactions with small changes in Hi-C ($|\log_2FC \text{ of normalized counts}| < 0.58$; i.e., $0.67 < FC < 1.5$) but substantial changes in ChIP intensity. We hypothesized that these differential interactions could be real enhancer-promoter interactions that are not captured with high sensitivity via Hi-C. Indeed, we find such differential HiChIP interactions with small changes in Hi-C but substantial changes in ChIP intensity and linked to a promoter are depleted in significant Hi-C interactions (odds ratio=0.71, Fisher’s exact test P value $< 1.0 \times 10^{-16}$) and enriched in differentially expressed genes (odds ratio=1.11, Fisher’s exact test P value $< 7 \times 10^{-3}$) compared to other HiChIP interactions linked to a promoter. IGV tracks in **Fig. R2** below and **Supplementary Fig. 22** show example loci where Hi-C interactions do not change between cell lines while there are substantial changes in ChIP-seq and gene expression along with HiChIP interactions. It seems that Hi-C interactions in both cell lines are enriched around subTAD boundaries, whereas HiChIP interactions can efficiently reveal cell line specific promoter anchored interactions for genes that are also exclusively expressed in that cell line.

Figure R2. IGV tracks of significant Hi-C and HiChIP interactions around the up-regulated genes in GM12878 including *LAMP3* and *KLHL6* showing HiChIP interactions (**bottom track**) change around this locus; while Hi-C interactions (**top track**) remain mostly stable. For gene expression, TPM values were illustrated as red being high, and gray being low. Gained differential ChIP-seq peaks in K562 over GM12878 were represented as red peaks, whereas lost ones were represented as blue.

6. What is the reproducibility of HiC-DC+ for detecting significant interactions and differential interactions?

For each pair of replicates of untreated and treated THP-1 Hi-C data, we generated scatter plots of Z-scores and $-\log_{10}(P)$ values for top 100000 Hi-C interactions in untreated and treated THP-1 and now added this as **Supplementary Fig. 2a-d**. We also generated the same plots for mESC H3K27ac HiChIP for top 100000 HiChIP interactions (**Supplementary Fig. 2e,f**). Scatter plots show that significance of interactions is in high agreement between replicates, suggesting high reproducibility both for HiC and HiChIP.

To test the reproducibility of differential interactions, we used Hi-C data with 4 replicates in mESC and NPC from Bonev et al., 2017. We found differential interactions between mESC and NPC using all three replicates (123), and all pairwise combinations of replicates (12, 13, and 23). We expect the same behavior with all these runs. The UpSetR plot that we included as **Supplementary Fig. 14** shows that only about 10% of differential interactions called using 2 replicates at a time are exclusive to individual pairs. (As expected, using all three replicates gives more power to detect interactions than any pair of replicates.)

7. Whether the HiC-DC+ results are consistent at different bin resolutions (for example, 25Kb, 10Kb and 5Kb)? Can it be applied to nucleosome resolution Micro-C data (PMID: 32213324)?

To check the consistency of HiC-DC+ interactions at different bin resolutions, we found the fraction of overlap between interactions called at two consecutive resolutions and added this analysis as **Supplementary Fig. 3**. ~80% of interactions called at 5kb, 25kb, and 50kb were also found as significant at 10kb, 50kb and 100kb, respectively.

HiC-DC+ package is very flexible in feature and statistical model definitions with the options to choose negative binomial or hurdle regression or either with distance dependent variable dispersion along with substantially improved efficiency. Thus, it readily offers the potential to be extended to other similar data sets. As requested, we tested HiC-DC+ on Micro-C (Krietenstein N et al., 2020; 4DNFI2TK7L2F) and CTCF ChIA-PET (4DNFIDCEC4PQ) utilizing negative binomial regression with variable dispersion at 5kb for Micro-C, and hurdle regression with distance dependence as power decay at 10kb as covariates for ChIA-PET. Our initial analyses show high overlap with HiCCUPS loops called on Hi-C data for both CTCF ChIA-PET (80%) and Micro-C (now included as **Supplementary Fig. 1**). We now mention this analysis when we discuss flexible modeling options in the **Results** and again in the **Discussion**.

8. When applying HiC-DC+ differential analysis to biological replicates of the same cell types, for examples, replicates of mESC Hi-C data (GSE96107) or replicates of mESC HiChIP data (GSE80820), all the identified differential interactions will be false positives. Can HiC-DC+ control type-I error in this scenario? Since HiC-DC+ identified much more differential interactions than other methods, I am concerned that HiC-DC+ results may contain high percentage of false positives.

Actually, HiC-DC+ called *fewer* such false positive differential interactions compared to other methods. To test whether HiC-DC+ introduces false positives, we conducted a differential analysis on 4 biological replicates of mES Hi-C from Bonev et al., 2017 by contrasting two dummy conditions containing 2 such replicates each. **Fig. R3** (below) demonstrates that only a handful of false positives are found in such a setting.

Figure R3. Volcano plot showing differential Hi-C interactions between two groups of biological replicates of mES Hi-C data from Bonev et al., 2017.

Minor comments:

1. In Fig S3, the red curve (HiC-DC+) and blue curve (FitHiChIP) look almost identical. I am not convinced that HiC-DC+ shows higher enrichment at TAD boundaries than FitHiChIP.

To illustrate this point better, we generated the CDF of enrichment of GM12878 SMC1A HiChIP signal at subTAD boundaries with respect to interior of subTADs by dividing the average value at subTAD boundaries by average values within the subTAD used to generate the metaplot in **Supplementary Fig. 11** and include it as **Supplementary Fig. 12**. Enrichment values for HiC-DC+ are found to be greater than that of FitHiChIP (P value $< 1 \times 10^{-16}$, Wilcoxon rank-sum test).

2. The differential analysis of HAP1 data is at 25Kb bin resolution (Fig 3). To better measure enhancer-promoter interactions, I hope they can perform such differential analysis at 5Kb and 10Kb bin resolution.

Our motivation behind the 25kb resolution choice was to identify differences in TAD reorganization, where low granularity is more appropriate, rather than detecting putative

enhancer-promoter interactions, that would warrant deeper libraries and higher resolutions. Nevertheless, we provide the results for 10kb below (**Fig. R4** below) that are directionally similar to those in the paper.

Figure R4. Differential interaction analysis between each TAD and its five flanking TADs. The barplots show the number of differential interactions in Δ WAPL vs wild type belonging to each category as identified by HiC-DC+ at FDR < 0.1.

3. The authors can provide more details on how they handle insertions, deletions and copy number variations in the cancer cell line K562. Those structural variations can introduce false positives in the identified significant interactions.

We agree that underlying copy number changes are a source of false positives, but addressing this problem is out of scope for the current paper as specialized algorithms are required to detect structural variants from Hi-C (see e.g. Wang et al., Genome Biol 2020 from the Park lab and Dixon et al., Nat Genet 2018 from the Xue lab).

4. As a new computational method, the authors need to provide more details on the computational cost (such as memory and running time), and compare that with other existing methods.

We summarized the computational cost for Hi-C and HiChIP data on different platforms and added this information as **Supplementary Tables 1-5**. Running the previous HiC-DC model with 8 cores in parallel on chromosome 1, 11 and 22 takes 13, 7, and 3 min and requires 14.2, 9, and 5 GB of memory, respectively.

Reviewer #2 (Expertise: Chromatin organization using HiC or HiChIP):

Sahin et al. created an update to HiC-DC called HiC-DC+. They evaluated the effectiveness of significant interaction identification in HiChIP data and the differential analysis of interactions from Hi-C and HiChIP experiments. The effort to integrate HiChIP interaction identification and statistical differential interaction calling is worthwhile and there are several packages that try to accomplish similar tasks that have been implemented since the creation of the original HiC-DC algorithm.

1. It was difficult to tell whether the program has significantly improved or not. The paper would be dramatically improved by a better description of why this update represents a significant advance in the field. From what I can tell, most of the analysis could have been done with the original HiC-DC algorithm in conjunction with DESeq2. The way the manuscript is written, this update seems to simply improve storage and parallelization and does not represent a departure from current methods. The key advances over the original algorithm should be made clearer.

The improvements were mentioned above (response to Reviewer #1, comment 1) with time and memory usage and can be found now in **Supplementary Tables 1-5**. We have also added an **Introduction** to the main text to highlight these major improvements of the HiC-DC+ package over the original HiC-DC method.

2. Using the CRISPRi-FlowFISH data to evaluate H3K27ac HiChIP interaction calling is a clever idea. However, HiChIP may display interactions that do not appear impactful in the CRISPRi-FlowFISH. This is because CRISPRi-FlowFISH is used to describe the impact that an enhancer has on gene expression. Interaction signal is only one piece of that puzzle as noted by the ABC model that the authors have cited. Thus, simply because CRISPRi-FlowFISH does not detect an impact on the gene's expression, does not mean that there was not an interaction.

We acknowledge this caveat: there could be a true H3K27ac interaction whose disruption has no detectable effect (at least by FlowFISH) on target gene expression; we have added this point to the **Discussion**. Nevertheless, given the lack of gold standard interaction calls, we find the CRISPRi-FlowFISH data set to be a useful benchmarking exercise.

3. In addition to CRISPRi-FlowFISH, the authors should use other methods to evaluate their significant HiChIP interactions. For example, they should include APA plots (average metaplots) and APA scores for interactions unique to HiC-DC+, those shared between algorithms, and those unique to other algorithms.

We generated separate APA plots for interactions that are unique to HiC-DC+ and FitHiChIP(L) and shared between HiC-DC+ and FitHiChIP(L) using mES H3K27ac and K562 H3K27ac HiChIP data (Mumbach et al., 2017) at 5kb resolution and included those plots as **Supplementary Fig. 7**. Our interactions have higher APA scores than FitHiChIP interactions for both data sets.

4. Differential HiChIP and differential Hi-C interactions should be evaluated in a similar fashion (APA plots in each sample and for unique vs shared differential interactions). This will allow the reader to evaluate how well each algorithm describes observable differences between maps.

We also generated APA plots for gained and lost Hi-C interactions in WAPL KO HAP1 cells at 25kb detected by HiC-DC+ and other differential interaction calling methods (diffHiC, multiHiCcompare and Selfish) and included those plots in **Supplementary Fig. 17-19**. Differential interactions found by HiC-DC+ have higher APA scores than other methods.

5. Line 147-148: “only Hi-C-DC+ recovered both findings: a majority of gained interactions connected neighboring TADs together with a drastic loss of intra-TAD interactions.”. It is unclear why the authors make this claim. Each of the other methods seem to show a similar effect.

The other methods only found one or the other findings with the exception of Selfish (and HiC-DC+). diffHiC finds more lost intra-TAD interactions than gained ones; however, it is not that dramatic. multiHiCcompare detects almost the same number of gained and lost intra-TAD interactions resulting in a slight net gain in intra-TAD interactions. Selfish has the correct overall properties but did poorly at recovering differential interactions at the 3 loci highlighted in the original paper.

6. I am very concerned that HiC-DC+ does not incorporate self-ligation / IP peak detection during HiChIP identification, particularly when thinking about differential interaction identification. Because HiChIP involves immunoprecipitation of interactions, any signal differences between samples can either be due to loss of interactions, or simply due to loss of immunoprecipitation efficiency. Thus, while the actual interactions within the cell may be unchanged, the HiChIP signal can be changed simply due to changes in the protein binding. Incorporating immunoprecipitation signal (ChIP-seq or self-ligations) is an important step to delineate interaction changes vs signal changes that are more ambiguous. It would be best to report both types of changes as distinct categories, but that requires estimating the IP efficiency of each locus. Without that step, the differential interaction calling is ambiguous.

We answered this question above in response to Reviewer #1, comment 5 (**Supplementary Fig. 21-22,27**, and **Fig. R2** above). In particular, in the majority of cases, we show that significantly differential HiChIP interactions are supported by differential Hi-C signal as well; indeed, in many cases, differential Hi-ChIP interactions involve a simultaneous and concordant change in Hi-C signal *and* ChIP-seq signal at one or both anchors. Moreover, when there are differential HiChIP interactions with only small changes in Hi-C, in some cases this appears to be due to a lack of sensitivity of Hi-C for finding promoter-enhancer interactions rather than simply a bias from differential ChIP signal. Therefore, we believe it is not so clear to decide on whether differential HiChIP interactions are “ambiguous” based only on ChIP (or self-ligation) signal. Indeed, we needed parallel Hi-C data to examine different possibilities. We now have added a **Supplementary Note 3** and **Results** section to the main text where some of these issues are addressed, since we believe this is a gap in the current literature.

7. The authors should explain why they see the ABD “outperforms the ABC score” (Supplementary Note). The cited ABC paper already performed a comparison to a model with distance (ABD) instead of contact (ABC) and found that the ABC model was better.

What we denote as ABD in our paper uses $1/\text{Distance}$ instead of Hi-C contacts C . In Fulco et al. (2019), authors “compared versions of the ABC model in which we estimated Contact for each DE–G pair...as a function of distance ($\text{Contact} \approx \text{Distance}^{-1}$)” and found an AUPRC of 0.64 for ABD, almost the same as their reported AUPRC of 0.65 for ABC (in their Figure 3a). The “Activity x $\text{Contact}_{\text{Fractal}}$ ” track on their Supplementary Figure 6a ($\text{Contact}_{\text{Fractal}}=1/\text{Distance}$ per their Supplementary Figure 6c) corresponding to ABD already has sections where the precision is greater than ABC for a given recall. We simply demonstrated the same result across genes in the subset of enhancer-promoter candidates (see **Methods**) used for benchmarking, namely that the genewise auPR values for ABD is greater than that of ABC.

8. The authors compare to differential loop calling in Phanstiel et al, which used HiCCUPS. This program was designed to identify punctate CTCF loops. Others have designed loop calling algorithms for a similar purpose (SIP, Rowley et al., Genome Res 2020 and Mustache, Ardakany et al., Genome Biology 2020). These only identified ~13,000, or ~18,000 loops in human cells. Yet the authors claim to have 64,844 differential loops. The types of interactions (CTCF loops vs enhancer-promoter interactions) that are identified by HiC-DC+ is therefore probably different from CTCF loop callers like HiCCUPS. If the intention is to call enhancer-promoter interactions, it should be clearly stated that the intention of these programs is quite different. Additionally, these HiC-DC+ interactions should not be contrasted with HiCCUPS or other callers where the intent is different.

We agree with the reviewer that HiCCUPS is quite conservative and best for detecting CTCF-mediated loops rather than promoter-enhancer interactions. However, the original Phanstiel et al. paper did use HiCCUPS, and one theme of the paper indeed was examination of differential regulatory interactions. Moreover, HiCCUPS is widely used in the community and even viewed as a gold standard method (see request from Reviewer #1 for overlapping HiC-DC+ and FitHiChIP interactions with HiCCUPS calls on parallel Hi-C data for regulatory interactions in the CRISPRi-FlowFISH comparison). We have added comments on the differences between structural loop and regulatory interaction calls in the **Discussion** to address these issues.

9. The authors should provide the number of total loops called by each method for each dataset, including each Hi-C and Hi-ChIP datasets. This is important to evaluate differences between algorithms.

We added this as **Supplementary Table 6**.

10. It is known that sequencing depth impacts Hi-C data analysis, particularly regarding feature identification and differential analysis. There is a trade-off between using replicates separately (as DESeq2 incorporation requires) vs pooling replicates to obtain deeper matrices. It would be

valuable for the authors to evaluate how sequencing depth affects HiChIP feature calling, and how it may affect differential interaction calling. On a side note, it would also be valuable to evaluate how imbalanced depth between two replicates could impact differential interaction calling.

To clarify, we do pool replicates to obtain deeper contact matrices in order to identify significant interactions in each cell type, which we combine into an atlas of events to test in the differential analysis by DESeq2. For the differential analysis, we then use the counts from each replicate together with distance-dependent scaling factors with DESeq2. We now explicitly describe this process in the text, since the full explanation was missing before. For the differential part of the analysis, we would not characterize the use of replicates as a trade-off with coverage, since there is no principled way to assess the significance of differential interactions without replicates – one could only obtain a fold change from the pooled data with no P value.

Sequencing depth in HiChIP, like Hi-C, affects the resolution (genomic bin size) at which one can do the analysis. We performed empirical downsampling analyses for Hi-C in our previous HiC-DC paper (Carty et al., 2017) to show how detection of interactions (based on full coverage interaction calls) degrades with lower coverage at the same bin size. On the other hand, we show in **Supplementary Fig. 3** that for very deep Hi-C data, changing the binning resolution by a factor of 2 still gives fairly consistent results. For differential interactions, the behavior will follow the theoretical rules of differential read count analysis, namely that higher sequencing coverage improves detection of differential events with relatively low read counts that are dominated by Poisson noise, while a greater number of replicates improves detection of differential events with higher read counts but more subtle effect sizes. With respect to unbalanced HiChIP replicates, similar to differential read count analysis for any assay (e.g. RNA-seq, ChIP-seq, etc.), library size scaling factors cannot correct for very large differences in coverage. We have only performed differential analyses where Hi-C or HiChIP replicates of similar coverage are available. Since these properties are not specific to HiChIP, we did not include additional empirical experiments, although we could add this information as discussion points if the reviewer feels it would be helpful.

11. I didn't see any of the usual benchmarking. How much memory does HiC-DC+ use for interaction calling in Hi-C, HiChIP, and differential interaction calling? How long does it take with various computing power?

We now provide time and memory usage in **Supplementary Tables 1-5**.

Reviewers' Comments:

Reviewer #1:

Remarks to the Author:

In this revision, the authors provided additional data to improve the manuscript. However, several of my previous comments have not been fully addressed. I have some serious concerns about their key conclusions. Here are my specific comments.

1. I still don't believe that HiC-DC+ achieved significant improvement over FitHiChIP. In Fig S5, if users care about a reasonable recall (>0.2), FitHiChIP achieved consistently higher precision than HiC-DC+. HiC-DC+ outperforms FitHiChIP when recall is <0.2 . I think HiC-DC+ results are biased towards interactions with strong H3K27ac ChIP enrichment, while missing interactions with median or low H3K27ac ChIP enrichment.

2. I have major concern about the authors claim on ChIP enrichment level. The authors used CRISPRi-FlowFISH data from 22 genes in K562 cells to benchmark interactions identified from different modeling approaches in Supplementary Note 1. I full agree with Reviewer 2's comment that "HiChIP may display interactions that do not appear impactful in the CRISPRi-FlowFISH". In particular, for H3K27ac HiChIP data, most identified interactions are enhancer-enhancer interactions, and only a small proportion of them are enhancer-promoter interactions. There are no benchmark data for the majority of enhancer-enhancer interactions, where HiC-DC+ results can contain strong ChIP enrichment level bias.

One way to evaluate the effect of ChIP enrichment level can be based on GM12878 Smc1a HiChIP data. The authors can use HiCCUPS loops from Rao et al 2014 paper as the gold standard, and evaluate the effect of different types of modeling of ChIP enrichment level.

3. Since the differential HiChIP interactions are driven by both ChIP signal and Hi-C interactions, if using the differential interactions identified from Hi-C data as the gold standard, I would expect that ignoring ChIP enrichment in HiChIP differential analysis will lead to biases. I feel it is difficult to follow how the authors evaluate ChIP enrichment bias in HiChIP differential analysis.

4. I disagree with the authors' interpretation on example in Figure R2. Hi-C data show that the interactions are shared between K562 and GM12878, and differential interactions identified from HiChIP data are driven by cell-type-specific H3K27ac peaks. Since the goal is to identify cell-type-specific enhancer-promoter interactions, the authors need to account for biases from ChIP signal.

5. Fig S2 shows high correlation between $-\log$ p-values. Since many dots are at lower left corner, it is still difficult to evaluate the actual overlap. I hope the authors can use the same thresholds to call interactions between two biological replicates, and show Venn diagram of overlap, similar to Figure 3B in Rao et al 2014 paper. I would expect 60~70% interactions are reproducible between replicates.

6. I think Reviewer 2 made an excellent suggestion of using APA plot. In Figure S7, in addition to use HiChIP data for APA, the authors also need to use K562 and mESC Hi-C data for APA, to evaluate enrichment of interactions without biases from ChIP signal.

Reviewer #2:

Remarks to the Author:

Overall, I appreciate the efforts of the authors to address the reviewers' concerns. The additions to the manuscript are helpful. However, a few issues could use some further clarification.

1. I thank the authors for adding APA plots as requested. However, the APA plots of differential interactions in WAPL KO cells don't address the main point of my original comment. The original intent

of the comment was to show the differences between CTL and WAPL KO cells. First, it's unclear from the figure legend which cell type's signal they are showing in these plots (Supp Fig 17-19). Second, the authors appear to only show the APA plots for one condition (WAPL KO cells?). They should show the APA plot of the CTL side-by-side with the WAPL KO cells for each loop category. Currently these APA plots seem to indicate that HiC-DC+ interactions "up in WAPL KO" and interactions "down in WAPL KO" have the same signal in WAPL KO cells (which is odd, but may be case), but the important point would be to directly allow the reader to compare the signal in CTL to signal in WAPL KO for each loop category.

2. HiChIP signal vs ChIP signal. The authors perform analysis with H3K27ac HiChIP and ChIP-seq showing that loss of HiChIP corresponds to loss of ChIP-seq and loss of Hi-C. It's unclear if this is specific to H3K27ac which seems to be important for enhancer function. Potential users of HiC-DC+ will be performing HiChIP with IP of other proteins. Therefore, it's problematic to draw conclusions about the relationship between Hi-C, HiChIP, and ChIP efficiencies from this one mark. I encourage the authors to add this caveat to their discussion.

Reviewers' comments:

Reviewer #1 (Remarks to the Author):

In this revision, the authors provided additional data to improve the manuscript. However, several of my previous comments have not been fully addressed. I have some serious concerns about their key conclusions. Here are my specific comments.

1. I still don't believe that HiC-DC+ achieved significant improvement over FitHiChIP. In Fig S5, if users care about a reasonable recall (>0.2), FitHiChIP achieved consistently higher precision than HiC-DC+. HiC-DC+ outperforms FitHiChIP when recall is <0.2. I think HiC-DC+ results are biased towards interactions with strong H3K27ac ChIP enrichment, while missing interactions with median or low H3K27ac ChIP enrichment.

For convenience, we include **Supplementary Fig. 5** below (**Fig. R1**). We are presenting the CRISPRi-FlowFISH data with a precision-recall curve because this is a difficult task where most tested candidate enhancers do not represent true functional enhancer-promoter interactions (i.e. many negatives, only a few positives). In this setting, as is widely understood in statistics and machine learning, one should compare the most confident predictions of all methods at the high precision end of the curve. (One can also consider the overall ranking by reporting the area under the precision-recall curve, where HiC-DC+ also outperforms FitHiChIP and other methods.) At a recall of 0.2, both HiC-DC+ and FitHiChIP have a precision of about 0.3. For recall >0.3, both methods have a poor precision under 0.2. It is not statistically appropriate to claim superiority of FitHiChIP at higher recall, when at this place in the ranking, the method is making 2 to 4 (or more) false positive predictions for every true positive prediction.

The reviewer asserts that FitHiChIP is performing above HiC-DC+ at the noisy high recall/poor precision end of the curve because "I think HiC-DC+ results are biased towards interactions with strong H3K27ac ChIP enrichment." To address this speculation, we examined the average normalized H3K27ac ChIP signal for the candidate pairs in Fulco et al., 2019 deemed significant by each method and found *no statistically significant difference* in ChIP signal for interactions at recall ≤ 0.2 (**Fig. R2** below, two-sided Wilcoxon test). This contradicts the reviewer's assertion that HiC-DC+ outperforms FitHiChIP on confident predictions due to strong ChIP enrichment. Moreover, FitHiChIP interactions overlapping the Fulco et al. candidate pairs at recall > 0.2 actually have *slightly but significantly higher ChIP signal* than the comparable low confidence HiC-DC+ interactions (**Fig. R2** below). This contradicts the suggestion that FitHiChIP is better able to find interactions with medium or low ChIP enrichments when considering lower confidence/higher recall predictions.

An alternative possibility is that biases of the FitHiChIP method align with biases of the CRISPRi-FlowFISH data set (e.g. enrichment for shorter genomic distance), giving FitHiChIP an advantage at the noisy end of the precision-recall curve. However, our main point here is that the advantage of HiC-DC+ for high confidence predictions cannot be explained by a bias towards high ChIP enrichment.

Figure R1 (Supplementary Fig. 5). Precision-recall curves for different methods on the data used to generate **Fig. 2a**.

Figure R2. Distribution of H3K27ac ChIP signal in candidate enhancer anchors of pairs found significant by either HiC-DC+ or FitHiChIP at recall values lower or higher than 0.2 (P values reported based on two-sided Wilcoxon tests).

2. I have major concern about the authors claim on ChIP enrichment level. The authors used CRISPRi-FlowFISH data from 22 genes in K562 cells to benchmark interactions identified from different modeling approaches in Supplementary Note 1. I full agree with Reviewer 2's comment that "HiChIP may display interactions that do not appear impactful in the CRISPRi-FlowFISH". In particular, for H3K27ac HiChIP data, most identified interactions are enhancer-enhancer interactions, and only a small proportion of them are enhancer-promoter interactions. There are no benchmark data for the majority of enhancer-enhancer interactions, where HiC-DC+ results can contain strong ChIP enrichment level

bias.

We acknowledge in the manuscript that CRISPRi-FlowFISH is an imperfect performance measure for H3K27ac HiChIP calls – as Reviewer #2 wrote, there can be true HiChIP interactions that for various reasons (e.g. compensation by other 3D interactions) do not affect target gene expression when perturbed by CRISPRi. We note however that the FitHiChIP manuscript also included CRISPRi data sets in order to validate their H3K27ac HiChIP calls (Bhattacharya et al., 2019). Importantly, if ChIP bias were truly a problem for our method, i.e. if HiC-DC+ were truly identifying *false positive HiChIP interactions* with high ChIP signal at one of the anchors, one would expect that HiC-DC+ would have a *high number of false positives* at the top of its ranking compared to methods that correct for ChIP bias. The CRISPRi-FlowFISH results show this is not true – rather, the methods that correct for ChIP bias suffer from false positives in their most confident predictions based on precision-recall analysis.

If on the other hand, the reviewer's claim is that HiC-DC+ is detecting *true positive HiChIP interactions* with a high associated H3K27ac signal that are most likely to have regulatory impact, we believe this would in fact support the usefulness of HiC-DC+. Experimentalists are performing H3K27ac HiChIP to enrich for regulatory interactions. Correcting for ChIP bias is not an end in itself and certainly should not *deplete* for true interactions associated with the ChIP signal of interest. As mentioned above, when examining high confidence interactions (recall ≤ 0.2) on the CRISPRi-FlowFISH data, there is actually no significant difference in ChIP enrichment between HiC-DC+ and FitHiChIP (**Fig. R2**). In fact, in **Fig. R3** below, we find that the top 100,000 interactions identified by FitHiChIP have higher average normalized ChIP signal levels in both anchors compared to HiC-DC+ interactions.

The statement that “HiC-DC+ results can contain strong ChIP enrichment level bias” for enhancer-enhancer interactions is speculation that cannot be addressed because, as the reviewer states, there is no benchmark for these interactions.

Figure R3. Distribution of averaged normalized H3K27ac ChIP signal of the top 100,000 interactions identified by HiC-DC+ and FitHiChIP in K562 H3K27ac HiChIP. ChIP signal in the (a) left anchor, (b) the right anchor, and (c) the product of signal in both anchor.

One way to evaluate the effect of ChIP enrichment level can be based on GM12878 Smc1a HiChIP data. The authors can use HiCCUPS loops from Rao et al 2014 paper as the gold

standard, and evaluate the effect of different types of modeling of ChIP enrichment level.

This is a new request not raised in the previous review. Nonetheless, we performed this analysis and show that using HiCCUPS loops on GM12878 as the ground truth, detection of these loops by SMC1A HiChIP in GM12878 *without modeling ChIP bias* gives very similar results to FitHiChIP *with modeling ChIP bias* (**Fig. R4**). While there might be some small advantage for FitHiChIP over HiC-DC+ on this task (506 vs. 481 events detected in the top 1000 predictions, 2422 vs 2293 events detected in the top 10000 predictions), this result demonstrates (i) there is no serious problem with ChIP enrichment bias in HiC-DC+ HiChIP calls and (ii) HiC-DC+ performs comparably to FitHiChIP *without requiring an additional ChIP-seq assay for normalization*.

Figure R4. Comparison of SMC1A HiChIP calls. **(Left)** Precision-recall curve for detection of HiCCUPS loops on GM12878 Hi-C (defined as ground truth) with HiC-DC+ applied to SMC1A HiChIP vs FitHiChIP applied to SMC1A HiChIP and SMC1A ChIP-seq in GM12878. **(Right)** Venn diagram of HiCCUPS calls, HiC-DC+ HiChIP calls at 1% FDR, and FitHiChIP calls at 1%FDR.

3. Since the differential HiChIP interactions are driven by both ChIP signal and Hi-C interactions, if using the differential interactions identified from Hi-C data as the gold standard, I would expect that ignoring ChIP enrichment in HiChIP differential analysis will lead to biases. I feel it is difficult to follow how the authors evaluate ChIP enrichment bias in HiChIP differential analysis.

We have included a detailed Supplementary Note explaining differential Hi-C and differential HiChIP analyses and have summarized the findings in the paper. While the reviewer expects there should be biases, our analyses do not find evidence for the claimed ChIP biases. Since the reviewer does not specify what part of the argument was difficult to follow, we did not revise these explanations. The key paragraphs in the paper are repeated below:

“We found that a majority of all differential HiChIP interactions called by HiC-DC+ – with no normalization for ChIP – were in fact supported by a concordant change in Hi-C

signal (58%, n=269,995) or in ChIP-seq signal at one or both anchors (70%, n=327,437) or concordant changes in both Hi-C and ChIP-seq signals (42%, n= 199,029) (**Supplementary Fig. 21 (now Supplementary Fig. 24)**). Most of the differential HiChIP interactions with a concordant change in ChIP-seq signal also exhibit a concordant change in Hi-C signal (61%), suggesting that the gain/loss of the ChIP signal most often happens simultaneously with change in the underlying 3D interaction, confounding the notion of normalizing for ChIP bias in HiChIP analysis.

We also identified differential HiChIP interactions with concordant differential ChIP signal in one of the anchors but only a small change in Hi-C signal (27%, n=128,408 of all differential HiChIP interactions). We hypothesized that these differential interactions included real enhancer-promoter interactions that are not captured with high sensitivity in Hi-C. Indeed, we found that differential HiChIP interactions linked to a promoter in one anchor with small changes in Hi-C but substantial changes in ChIP intensity in the other anchor are depleted in significant Hi-C interactions (odds ratio=0.71, $P < 1.0 \times 10^{-16}$, Fisher's exact test) but enriched in differentially expressed genes (odds ratio=1.11, $P < 7 \times 10^{-3}$, Fisher's exact test) compared to other promoter-anchored HiChIP interactions, suggesting that Hi-C lacked the sensitivity to detect these promoter-anchored interactions in either cell type, at least at the sequencing depth of the available experiments (**Supplementary Figure 22 (now Supplementary Fig. 25)**).

In addition to these empirical analyses that support HiC-DC+'s choice not to normalize for ChIP bias, there may also be a theoretical statistical reason to avoid ChIP normalization. We believe that ChIP normalization in HiChIP analysis may be analogous to conditioning on a post-treatment variable in estimates of causal effects of a treatment, especially in the context of differential analyses where the cell type can be viewed as a treatment (and ChIP-seq signal as a post-treatment variable). See e.g. "How Conditioning on Posttreatment Variables Can Ruin Your Experiment and What to Do about It", Montgomery et al., *Am J Pol Sci* 2018, for a recent presentation of this concept in a non-biological context; importantly, conditioning on the post-treatment variable can cause bias of any size and in either direction in estimating the causal effects of the treatment. We do not make this argument in the paper as it would need further statistical development in the setting of background model estimation. However, we mention this reference because it is not at all clear that the current practice of normalizing for ChIP signal in HiChIP analyses – while well-intentioned and trying to err on the side of being conservative – is statistically justified.

4. I disagree with the authors' interpretation on example in Figure R2. Hi-C data show that the interactions are shared between K562 and GM12878, and differential interactions identified from HiChIP data are driven by cell-type-specific H3K27ac peaks. Since the goal is to identify cell- type-specific enhancer-promoter interactions, the authors need to account for biases from ChIP signal.

Experimentalists perform H3K27ac HiChIP to identify regulatory interactions that could only be seen in Hi-C at extremely high coverage. We show specific examples where Hi-C only detects TAD-like interactions in both cell types, while HiC-DC+ HiChIP calls identify enhancer-promoter loops associated with upregulation of target gene expression. We also show this signal (association of differential enhancer-promoter HiChIP interactions with differential target gene expression) is true globally for cases where the Hi-C signal has no significant change.

Biologically, cell-type specific enhancer-promoter loops can indeed arise from cell-type specific enhancer accessibility/acetylation. In particular, we direct the reviewer's attention to the THP-1 monocyte to macrophage differential Hi-C analysis at the end of the paper, where with very deep Hi-C data (~5B read pairs), we do see gained Hi-C promoter-

enhancer interactions associated with gain of accessibility and acetylation at induced genes. Very high resolution Hi-C or other higher resolution assays (Micro-C, 4C etc.) would therefore be needed to provide independent evidence that these (biologically reasonable) HiChIP interactions associated with target gene upregulation are real. However, in the absence of additional data, it is also not possible to dismiss these interactions as “ChIP bias”.

5. Fig S2 shows high correlation between $-\log p$ -values. Since many dots are at lower left corner, it is still difficult to evaluate the actual overlap. I hope the authors can use the same thresholds to call interactions between two biological replicates, and show Venn diagram of overlap, similar to Figure 3B in Rao et al 2014 paper. I would expect 60~70% interactions are reproducible between replicates.

The Fig. R5 below shows the Venn diagrams for untreated and treated THP-1 Hi-C data, as well as mES and GM12878 H3K27ac HiChIP. At 5kb resolution, we found that 80%, 81%, 63% and 87% of interactions identified in the respective primary replicates of untreated THP-1, treated THP-1 Hi-C, mES and GM12878 H3K37ac HiChIP were also called in the secondary replicates, reflecting high reproducibility. We now mention these overlaps in the caption of **Supplementary Fig. 2**.

Figure R5. Overlap of significant interactions found at FDR < 0.01 in each replicate of (a) untreated THP-1, (b) treated THP-1, (c) mES H3K27ac HiChIP, and (d) GM12878 H3K27ac HiChIP.

6. I think Reviewer 2 made an excellent suggestion of using APA plot. In Figure S7, in addition to use HiChIP data for APA, the authors also need to use K562 and mESC Hi-C data for APA, to evaluate enrichment of interactions without biases from ChIP signal.

We revised APA plots (in **Supplementary Fig. 7**) by using Hi-C maps instead of HiChIP maps now. We performed APA plots for the top 5,000 interactions. We note that APA plots are essentially 2D “meta- peak” visualizations and therefore are useful for sanity checks but do not assess statistical significance. Nevertheless, the APA plots show the desired

behavior for HiC-DC+ exclusive HiChIP interactions, in particular, HiC-DC+ exclusive interactions had higher APA scores than FitHiChIP-exclusive interactions.

Reviewer #2 (Remarks to the Author):

Overall, I appreciate the efforts of the authors to address the reviewers' concerns. The additions to the manuscript are helpful. However, a few issues could use some further clarification.

1. I thank the authors for adding APA plots as requested. However, the APA plots of differential interactions in WAPL KO cells don't address the main point of my original comment. The original intent of the comment was to show the differences between CTL and WAPL KO cells. First, it's unclear from the figure legend which cell type's signal they are showing in these plots (Supp Fig 17-19). Second, the authors appear to only show the APA plots for one condition (WAPL KO cells?). They should show the APA plot of the CTL side-by-side with the WAPL KO cells for each loop category. Currently these APA plots seem to indicate that HiC-DC+ interactions "up in WAPL KO" and interactions "down in WAPL KO" have the same signal in WAPL KO cells (which is odd, but may be case), but the important point would be to directly allow the reader to compare the signal in CTL to signal in WAPL KO for each loop category.

We apologize that we did not previously include all the requested data or label the plots clearly enough. We have updated the APA plots in **Supplementary Fig. 17, 18, and 19 (now Supplementary Fig. 20, 21, 22)** to address these requests. As observed in the APA plots between HAP1 control cells (**bottom two rows of the figures**) and HAP1 WAPL KO cells (**top two rows of the figures**), pixel intensity for interactions that weaken upon WAPL KO are highest in HAP1 control as compared to HAP1 WAPL KO, while pixel intensity for interactions that strengthen upon WAPL KO are highest in WAPL KO as compared to HAP1 control.

HiChIP signal vs ChIP signal. The authors perform analysis with H3K27ac HiChIP and ChIP-seq showing that loss of HiChIP corresponds to loss of ChIP-seq and loss of Hi-C. It's unclear if this is specific to H3K27ac which seems to be important for enhancer function. Potential users of HiC-DC+ will be performing HiChIP with IP of other proteins. Therefore, it's problematic to draw conclusions about the relationship between Hi-C, HiChIP, and ChIP efficiencies from this one mark. I encourage the authors to add this caveat to their discussion.

We have added this caveat to our discussion. We present these analyses because of the focus on regulatory interactions in our paper and because of the availability of H3K27ac ChIP-seq and HiChIP in multiple cell types.

Note for both reviewers:

We have updated **Supplementary Fig. 6, 7** (APA plots) and the number of loops in **Supplementary Table 6** as we realized that the Fulco et al. mES data was mapped to mm9.

Reviewers' Comments:

Reviewer #1:

Remarks to the Author:

The authors did great job in the revision and fully addressed most of my comments and suggestions. The manuscript has been significantly improved. I appreciate the authors' efforts during the revision, and just have one minor comment.

Related to my previous comment #2, most identified interactions from H3K27ac HiChIP data are enhancer-enhancer interactions. I understand that there is no gold standard of enhancer-enhancer interactions, I just hope the authors can comment on some potential alternatives to benchmark them, such as enhancer-enhancer interactions measured by orthogonal technologies, and provide general guidance for practitioners.

Reviewer #2:

Remarks to the Author:

Overall I believe HiC-DC+ shows promise and I think it's important to have new alternatives to Hi-C and HiChIP processing algorithms. However, I still have some concerns that were not fully addressed. I agree with Reviewer 1 that the Precision-Recall curves make it appear that FitHiChIP outperforms HiC-DC+. I disagree with the authors' response that we should focus on the left side of the graph (high precision). The reason to even generate the curve is to judge the tradeoff between precision and recall. In fact, researchers will want to identify as many interactions as possible and not just the most stringent of calls.

Similarly, the new Figure R4 appears to confirm that FitHiChIP outperforms HiC-DC+.

I also disagree with the statement: "Importantly, if ChIP bias were truly a problem for our method, i.e. if HiC-DC+ were truly identifying false positive HiChIP interactions with high ChIP signal at one of the anchors, one would expect that HiC-DC+ would have a high number of false positives at the top of its ranking compared to methods that correct for ChIP bias. The CRISPRi-FlowFISH results show this is not true.."

The original CRISPRi-FlowFISH paper (Engreitz lab) showed in their ABC model that Activity x Contact are important parameters to consider. Indeed sometimes this "Contact" is due to polymer distance if the Activity is high. Therefore, a true positive in the CRISPRi-FlowFISH could result simply from high Activity (in this case a bias in H3K27ac ChIP) x (essentially linear/random) Contact. Therefore, I don't think the CRISPRi-FlowFISH argument is suited to rebuttal to the reviewers' concerns.

Unfortunately, the added APA plots indicate that there may be some issues with differential calling. For example, while the loops "up in KO" are dramatic (no loop in WT) the decreased loops are not as obvious. Instead, it appears that the "Down in KO" loops have more dramatic changes to the background signal than to the loop itself. While the APA scores decrease, this could instead reflect a change in background signal.

Additionally, comparing Supplementary Fig 21 d vs j, it appears that the "Down in WAPL" is simply because the loop is being called 1-2 pixels off from each other between maps. This indicates that the differences in the blurriness of the Hi-C matrix between experiments could potentially impact the differential analysis.

Response to reviews

Reviewer #1 (Remarks to the Author):

The authors did great job in the revision and fully addressed most of my comments and suggestions. The manuscript has been significantly improved. I appreciate the authors' efforts during the revision, and just have one minor comment.

Related to my previous comment #2, most identified interactions from H3K27ac HiChIP data are enhancer-enhancer interactions. I understand that there is no gold standard of enhancer-enhancer interactions, I just hope the authors can comment on some potential alternatives to benchmark them, such as enhancer-enhancer interactions measured by orthogonal technologies, and provide general guidance for practitioners.

We have acknowledged the limitation of the lack of gold standard for enhancer-enhancer interactions in the Discussion and suggested some potential emerging data sources that might enable validation in the future:

“Many interactions detected in H3K27ac HiChIP data are putative enhancer-enhancer interactions, for which there is no gold standard data set for evaluation. Higher-resolution assays like HiCAR and Micro-C appear to enrich for shorter range regulatory interactions, so a careful analysis of these chromatin conformation assays together with parallel H3K27ac ChIP-seq could establish a curated set of enhancer-enhancer interactions on which to benchmark H3K27ac HiChIP calls. One could also try to validate HiChIP-based interactions involving specific enhancers using independent 4C experiments with the enhancer locus as viewpoint. Still, such analyses depend on statistical analysis of high-throughput 3C-based experiments and are therefore not entirely orthogonal to HiChIP. Moreover, we would ultimately want a functional validation for enhancer-enhancer interactions, and for this we would need a better understanding of their regulatory role. For example, one might perform CRISPRi experiments that target one enhancer participating in the putative enhancer-enhancer interaction and perform H3K27ac qPCR at the other enhancer to assess if there is distal loss of activity. Data sets of this nature may emerge in the next few years.”

Reviewer #2 (Remarks to the Author):

Overall I believe HiC-DC+ shows promise and I think it's important to have new alternatives to Hi-C and HiChIP processing algorithms. However, I still have some concerns that were not fully addressed.

I agree with Reviewer 1 that the Precision-Recall curves make it appear that FitHiChIP outperforms HiC-DC+. I disagree with the authors' response that we should focus on the left side of the graph (high precision). The reason to even generate the curve is to judge the tradeoff between precision and recall. In fact, researchers will want to identify as many interactions as possible and not just the most stringent of calls.

We agree that the precision-recall (PR) curve allows a comparison of the full ranking of predictions of two methods, but then the appropriate statistic to use to compare rankings is the area under the PR curve (auPR). As we reported, HiC-DC+ outperforms FitHiChIP by auPR (0.205 vs. 0.18) on the CRISPRi-FlowFISH task. While we agree that researchers want to identify as many interactions as possible, as we mentioned in the last review, a threshold at which the precision is 0.2—i.e. of all the predicted interactions, 20% are true positives and 80% are false positives—is not a trade-off any researcher would want to make. We therefore

respectfully stand behind all the statements made in the previous review about the appropriate interpretation of precision-recall curves.

Similarly, the new Figure R4 appears to confirm that FitHiChIP outperforms HiC-DC+.

FitHiChIP does have a small improvement over HiC-DC+ for SMC1A HiChIP analysis in GM12878 as evaluated by detection of loops called by HiCCUPS on parallel Hi-C data in GM12878 (auPR 0.244 for FitHiChIP vs. auPR 0.231 for HiC-DC+, 506 vs. 481 events detected in the top 1000 predictions, 2422 vs 2293 events detected in the top 10000 predictions). The key point here is that *HiC-DC+ does not require an additional SMC1A ChIP-seq experiment in order to normalize for ChIP bias*. Therefore, there is a different kind of trade-off, namely that the researcher does not need to perform a parallel ChIP experiment to do the 3D interaction analysis, and the HiC-DC+ precision-recall curve tracks that of FitHiChIP quite closely, yielding very similar performance.

I also disagree with the statement: "Importantly, if ChIP bias were truly a problem for our method, i.e. if HiC-DC+ were truly identifying false positive HiChIP interactions with high ChIP signal at one of the anchors, one would expect that HiC-DC+ would have a high number of false positives at the top of its ranking compared to methods that correct for ChIP bias. The CRISPRi-FlowFISH results show this is not true.."

The original CRISPRi-FlowFISH paper (Engreitz lab) showed in their ABC model that Activity x Contact are important parameters to consider. Indeed sometimes this "Contact" is due to polymer distance if the Activity is high. Therefore, a true positive in the CRISPRi-FlowFISH could result simply from high Activity (in this case a bias in H3K27ac ChIP) x (essentially linear/random) Contact. Therefore, I don't think the CRISPRi-FlowFISH argument is suited to rebuttal to the reviewers' concerns.

To address this comment, it is important to distinguish between *true positive* and *true negative* enhancer-promoter interactions as defined in the CRISPRi-FlowFISH assay, versus how the ABC model is able to successfully make its predictions.

The definition of a *positive* enhancer in the CRISPRi-FlowFISH assay is one where CRISPR interference with the candidate enhancer results in significant differential expression of the target gene based on RNA FISH. Please note that all candidate enhancers annotated as falling in promoter regions in the Engreitz study were removed, as were candidates within 5Kb away from the transcription start site (see **Methods**). Therefore, we hope the reviewer agrees that the *true positives* in the CRISPRi-FlowFISH analysis are indeed non-promoter elements with a functional impact on gene expression—and therefore they should also have a true H3K27ac-associated loop to the promoter—while the *true negatives* are those whose perturbation does not have a functional impact on gene expression and are unlikely to have an H3K27ac-associated loop to the promoter (we have acknowledged this latter assumption is imperfect in the **Discussion**). Some of these true negatives do have high H3K27ac signal (e.g. they could be enhancers of other genes). Therefore, the statement made in our previous response—*"if HiC-DC+ were truly identifying false positive HiChIP interactions with high ChIP signal at one of the anchors, one would expect that HiC-DC+ would have a high number of false positives at the top of its ranking"*—still stands. There is a set of true negative candidate elements with high ChIP signal that theoretically poses a challenge to our method because we do not correct for ChIP bias, and yet we do not see a performance hit relative to other methods. Please also note that although we do not model ChIP bias explicitly, we do use covariates that account for sources of systematic sequencing biases, such as GC content and mappability, and this may already be implicitly modeling a component of the ChIP bias.

The reviewer makes the statement that “a true positive in the CRISPRi-FlowFISH could result simply from high Activity” in the case that “[Contact is] due to polymer distance”. We interpret this statement to mean that the ABC model is sometimes able to correctly predict a *true positive* enhancer-promoter interaction on the basis of ChIP-signal (and accessibility, i.e. “Activity”) and proximity (linear genomic distance) to the promoter. We indeed make the same point in the paper, i.e. “Activity-by-Distance” outperforms ABC and in fact all the HiChIP methods (see **Supplementary Note 2**). Therefore, the reviewer’s concern is that ChIP bias is helping HiC-DC+ accurately detect *true positive* enhancer-promoter interactions. We addressed these comments at length in response to Reviewer #1, who has accepted our arguments. In particular, we showed that there is actually no significant difference in the ChIP signal between the high confidence interactions (recall < 0.2) predicted by HiC-DC+ (no ChIP bias correction) versus FitHiChIP (ChIP bias correction) (**Supplementary Fig. 9**).

We have already acknowledged the shortcomings of the CRISPRi-FlowFISH data set in the **Discussion**, including the paucity of true long-range interactions. To try to address the Reviewer #2’s remaining concern, we have added the following statement: “Finally, when comparing against HiChIP methods that correct for ChIP-seq bias, there is the confounding issue that strong H3K27ac ChIP-seq signal at accessible elements is associated with true functional enhancer-promoter interactions with detectable effect size in CRISPRi-FlowFISH, especially at close proximity to the promoter, as shown by Activity-by-Contact and Activity-by-Distance analysis (**Supplementary Fig. 11, Supplementary Note 2**). This raises the possibility that HiC-DC+, by not normalizing for ChIP-seq signal, is implicitly using ChIP bias to perform well in the CRISPRi-FlowFISH task. However, there is in fact no significant difference in ChIP signal between the high confidence interactions predicted by HiC-DC+ versus FitHiChIP, a method that does correct for ChIP bias (**Supplementary Fig. 9**). Since the goal of H3K27ac HiChIP is to enrich for regulatory interactions, including enhancer-promoter interactions that are expected to influence target gene regulation, we believe CRISPRi-FlowFISH is a useful benchmark data set for HiChIP interaction callers, despite the limitations described here.”

Unfortunately, the added APA plots indicate that there may be some issues with differential calling. For example, while the loops "up in KO" are dramatic (no loop in WT) the decreased loops are not as obvious. Instead, it appears that the "Down in KO" loops have more dramatic changes to the background signal than to the loop itself. While the APA scores decrease, this could instead reflect a change in background signal.

Additionally, comparing Supplementary Fig 21 d vs j, it appears that the "Down in WAPL" is simply because the loop is being called 1-2 pixels off from each other between maps. This indicates that the differences in the blurriness of the Hi-C matrix between experiments could potentially impact the differential analysis.

We believe that the difference in APA plots between the dramatic “up in KO” loops and more subtle “down in KO” reflects the true biology of the WAPL perturbation. That is, WAPL knockout leads to TAD aggregation, with strong new interactions between boundaries of nearby TADs, leading to dramatic “up in KO” loops visible in APA plots for KO where there was no loop in WT cells. However, WAPL knockout also leads to be more subtle reduction in intra-TAD interactions in the same TADs that have intensified “corner” interactions, i.e. strengthened interactions between the TAD’s boundary elements. This accumulation of “corner” interactions will sometimes occur close enough in the contact map to the lost intra-TAD interaction to be captured in the APA plot, leading an overall increase in blurriness and/or background signal in KO as compared to WT for the “down in KO” interactions. Note that almost half of the HiC-DC+ detected “down in KO” loops do not exhibit this blurriness as can be seen in **Supplementary**

Fig. 21e,k. In addition, there is another detailed section on our differential Hi-C analysis for THP-1 monocyte to macrophage differentiation, where e.g. differential enhancer-promoter interactions are validated by concordant changes in accessibility, H3K27ac modification, and gene expression. This provides evidence orthogonal to APA analysis that HiC-DC+ is identifying meaningful differential interactions.